# Learning Structure-Semantic Evolution Trajectories for Graph Domain Adaptation

**Wei Chen[1],** **Xingyu Guo[1],** **Shuang Li[1],** **Yan Zhong[2],** **Zhao Zhang[3],**

**Fuzhen Zhuang[1,4,†],** **Hongrui Liu[5],** **Libang Zhang[5],** **Guo Ye[5],** **Huimei He[5]**

[1]School of Artificial Intelligence, Beihang University, Beijing, China
[2]School of Mathematical Sciences, Peking University, Beijing, China
[3]School of Computer Science and Engineering, Beihang University, Beijing, China
[4]Zhongguancun Laboratory, Beijng, China
[5]Independent Researcher, Beijing, China
{chenwei23,shuangliai,zhuangfuzhen}@buaa.edu.cn

## Abstract

Graph Domain Adaptation (GDA) aims to bridge distribution shifts between domains by transferring knowledge from well-labeled source graphs to given unlabeled target graphs. One promising recent approach addresses graph transfer by discretizing the adaptation process, typically through the construction of intermediate graphs or stepwise alignment procedures. However, such discrete strategies often fail in real-world scenarios, where graph structures evolve continuously and nonlinearly, making it difficult for fixed-step alignment to approximate the actual transformation process. To address these limitations, we propose **DiffGDA**, a **Diff**usion-based **GDA** method that models the domain adaptation process as a continuous-time generative process. We formulate the evolution from source to target graphs using stochastic differential equations (SDEs), enabling the joint modeling of structural and semantic transitions. To guide this evolution, a domain-aware network is introduced to steer the generative process toward the target domain, encouraging the diffusion trajectory to follow an optimal adaptation path. We theoretically show that the diffusion process converges to the optimal solution bridging the source and target domains in the latent space. Extensive experiments on 14 graph transfer tasks across 8 real-world datasets demonstrate DiffGDA consistently outperforms state-of-the-art baselines. Code is available at DiffGDA.

## 1 Introduction

Graph Neural Networks (GNNs) (Zhu et al., 2021; Wu et al., 2022) have demonstrated remarkable success in various real-world applications (Zhang et al., 2022a; Therrien et al., 2025; Chen et al., 2024; 2025b). However, their performance often degrades under distribution shifts across training and testing graphs (Liu et al., 2020; Qin et al., 2022). To overcome this limitation, Graph Domain Adaptation (GDA) aims to transfer knowledge from a labeled source graph to an unlabeled target graph by mitigating both feature distribution discrepancy and structural divergence (Dai et al., 2022; Liu et al., 2024a; Chen et al., 2025a). This work focuses on node-level GDA (Ding et al., 2018; Wang et al., 2024), tackling critical challenges posed by feature and topological shifts across graphs.

In recent years, a wide range of methods have emerged to enhance GDA, which can be broadly categorized into two paradigms (Huang et al., 2024): **(1) Model-oriented** approaches (Xiao et al., 2023; Pilancı & Vural, 2020; Dan et al., 2024; Fang et al., 2025b) aim to learn domain-invariant representations by aligning both semantic and structural information through objectives such as minimizing subgraph structural distances (Liu et al., 2023a; You et al., 2023), reducing Maximum Mean Discrepancy (MMD) (Chen et al., 2019; 2025a), and using adversarial discriminators (Dai et al., 2022). While these methods promote representation sharing, they assume limited structural variation between domains, which restricts their effectiveness when substantial structural differences exist (Huang et al., 2024). **(2) Data-oriented** methods (Na et al., 2021; Huang et al., 2024; Lei

---

[†]Corresponding author.

et al., 2025) address these challenges by constructing intermediate graphs to bridge structural gaps, projecting both domains into a shared semantic space. This facilitates more stable training by aligning structures and capturing cross-graph correspondences, offering a more flexible and interpretable framework for GDA, and showing promising potential in tackling complex graph domain shifts.

However, most data-oriented methods rely on the assumption that **the graph transfer process is discrete**, meaning the source graph can be transformed into the target graph through a limited number of alignment steps. Unfortunately, this assumption often fails in real-world scenarios, particularly when dealing with unlabeled graphs, due to the key limitations: graphs are governed by different underlying processes (Deng et al., 2019; Sharma et al., 2024), such as social dynamics, citation growth, or knowledge diffusion, which evolve nonlinearly and in domain-specific ways. Their evolution is driven by complex, context-dependent dynamics, rather than fixed or proportional structural patterns. Additionally, semantic information arises from the interplay between node features and structural context, but it often lacks explicit anchors for alignment (Heimann et al., 2018; Zhao et al., 2023). These factors make it difficult to align graphs with a fixed number of steps, as structural and semantic differences cannot be effectively bridged through simple alignment processes.

Along this research line, **continuous evolution** (Zeng et al., 2024; Zhang et al., 2021) provides a more natural modeling paradigm for graph transfer. Rather than manually constructing discrete intermediate graphs, this approach directly models the underlying evolution process, where the source graph gradually transforms into the target graph over continuous time. This paradigm not only bypasses the limitations of step-by-step alignment but also offers several advantages for GDA: (1) Structural changes are represented as smooth temporal trajectories, enabling flexible adaptation to nonlinear and heterogeneous topologies without relying on rigid graph (or subgraph) level alignment. (2) Semantic information evolves continuously along the transformation path, allowing the model to automatically learn optimal alignment trajectories and establish accurate cross-domain correspondences in an end-to-end manner. This perspective motivates a generative view of GDA, where the transfer of structure and semantics is unified as a time-driven evolutionary process. Inspired by the success of diffusion models in capturing complex distributional transformations (Croitoru et al., 2023; Zeng et al., 2024), we adopt a stochastic differential equation (SDE) (Protter, 2012; Li et al., 2020) framework to represent cross-graph transfer as a continuous probabilistic flow.

In this work, we propose **DiffGDA**, a **Diff**usion-based **GDA** that models both structural and semantic adaptation as a continuous-time evolution process. To guide this evolution, we introduce a domain-aware network that learns to steer the generative trajectory toward the target domain by uncovering the optimal adaptation path. This adaptive guidance allows the model to overcome the limitations of conventional discrete alignment methods by continuously directing the evolution toward the target domain, resulting in more coherent structural alignment and semantic adaptation. Furthermore, we provide a theoretical guarantee that the guided diffusion process converges to an optimal adaptation trajectory, effectively bridging the source and target domains.

Overall, our contributions can be summarized as:

(i) **Continuous Evolution Modeling.** We formulate GDA as a continuous-time generative process, jointly capturing structural and semantic transitions via stochastic differential equations. To the best of our knowledge, DiffGDA is the first work that brings diffusion into GDA. (ii) **Domain-Aware Guidance Network.** We propose a density-ratio-based guidance mechanism that adaptively steers the diffusion trajectory, enabling precise, target-aware evolution. (iii) **Empirical Superiority.** Extensive experiments on 14 real-world transferring tasks demonstrate the effectiveness of the proposed DiffGDA, consistently outperforming state-of-the-art baselines.

## 2 RELATED WORK

**Graph Domain Adaptation (GDA)** has attracted significant research attention due to its crucial role in enabling knowledge transfer between graphs with distributional shifts (Ding et al., 2018; Zhang et al., 2022b; Liu et al., 2023a; 2024a; Chen et al., 2025a; Guo et al., 2025; Kou et al., 2025b;a). Early approaches (Zhang et al., 2019; Wu et al., 2020; Dai et al., 2022; Pilancı & Vural, 2020; Dan et al., 2024) primarily adopted a model-driven paradigm, focusing on learning domain-invariant representations through techniques like adversarial training and MMD minimization (Liu et al., 2024a; You et al., 2023). While effective, these methods often struggle due to their reliance on rigid parametric

frameworks, which impose inherent limitations on their performance. Recently, data-driven methods (Huang et al., 2024; Lei et al., 2025) have emerged as a promising alternative, addressing domain gaps through intermediate graph construction and transformation. However, these approaches typically rely on heuristic alignment rules, which often lack generality and cannot adaptively discover or generate intermediate graphs that best suit the structural and semantic characteristics of the graph data, thereby limiting their effectiveness in handling complex graph domain transitions.

**Graph Diffusion Models.** Diffusion models (Cao et al., 2024) have achieved significant success across a variety of domains (Kong et al., 2021; Saharia et al., 2022; Luo et al., 2024b; Zhang et al., 2025; Chen et al., 2025c). In graph domain, these models are increasingly applied to model complex graph-structured data, particularly for tasks such as node and edge generation, molecule creation, and protein design (Niu et al., 2020; Jo et al., 2022; Konstantin Haefeli et al., 2022). By carefully designing the noising process and model architectures, symmetry constraints on the transition kernel and prior distribution can be explicitly incorporated, ensuring that the generated data follows roto-translationally invariant distributions. Recent advancements have expanded these frameworks beyond score-based formulations, introducing techniques like flow matching (Lipman et al., 2023; Liu et al., 2023b) and optimal transport-based diffusion (Peyré et al., 2019). These approaches provide alternative mechanisms to enforce geometric and permutation symmetry constraints in graph generation tasks. However, most existing methods have focused on symmetric diffusion processes. This presents a key challenge for diffusion-based domain adaptation models: how to leverage asymmetric diffusion mechanisms to enable transfer learning across heterogeneous graphs.

## 3 PRELIMINARIES

**Notation.** Given a labeled source graph $\mathbf{G}^{\mathcal{S}} = (\mathcal{V}^{\mathcal{S}}, \mathcal{E}^{\mathcal{S}})$, where $\mathcal{V}^{\mathcal{S}}$ is the set of nodes, $\mathcal{E}^{\mathcal{S}} \subseteq \mathcal{V}^{\mathcal{S}} \times \mathcal{V}^{\mathcal{S}}$ is the set of edges, and the nodes $\mathcal{V}^{\mathcal{S}}$ are associated with a feature matrix $\mathbf{X}^{\mathcal{S}} \in \mathbb{R}^{N_{\mathcal{S}} \times F}$ and a label matrix $\mathbf{Y}^{\mathcal{S}} \in \{0,1\}^{N_{\mathcal{S}} \times C}$. The adjacency matrix of the source graph is denoted as $\mathbf{A}^{\mathcal{S}} \in \{0,1\}^{N_{\mathcal{S}} \times N_{\mathcal{S}}}$. Similarly, we have an unlabeled target graph $\mathbf{G}^{\mathcal{T}} = (\mathcal{V}^{\mathcal{T}}, \mathcal{E}^{\mathcal{T}})$ with node features $\mathbf{X}^{\mathcal{T}} \in \mathbb{R}^{N_{\mathcal{T}} \times F}$ and adjacency matrix $\mathbf{A}^{\mathcal{T}} \in \{0,1\}^{N_{\mathcal{T}} \times N_{\mathcal{T}}}$. Both graphs share the same feature space ($F$-dimensional) and label space ($C$ classes), but they exhibit domain shift. In this work, our goal is to train a GNN that accurately predicts the node labels of the target graph $\mathbf{G}^{\mathcal{T}}$.

**Diffusion Models.** Diffusion models (Croitoru et al., 2023; Cao et al., 2024) have emerged as leading generative techniques in various fields (Luo et al., 2024b; Kong et al., 2021; Liu et al., 2025). These models are built upon the concept of a diffusion process that gradually transforms data into noise and then trains models to reverse this process using a well-defined objective function. Formally, the forward process incrementally corrupts data $\mathbf{X}_0 \sim p(\mathbf{X}_0)$ into noise through a continuous-time stochastic differential equation (SDE) (Protter, 2012):

$$\mathrm{d}\mathbf{X}_t = \mathbf{f}(\mathbf{X}_t, t)\mathrm{d}t + g(t)\mathrm{d}\mathbf{w}_t, \quad t \in [0, \mathsf{T}], \tag{1}$$

where $\mathbf{w}_t$ denotes the standard Brownian motion (also called the Wiener process) (Luo et al., 2024a), $\mathbf{f}(\mathbf{X}_t, t) : \mathbb{R}^d \times [0, \mathsf{T}] \to \mathbb{R}^d$ is a vector-valued drift coefficient determining the deterministic evolution of the system, and $g(t) : [0, \mathsf{T}] \to \mathbb{R}$ is a scalar diffusion coefficient controlling the magnitude of Gaussian noise added at each time step. A canonical example is the Ornstein-Uhlenbeck process (Maller et al., 2009), where $\mathbf{f}(\mathbf{X}_t, t) = -\frac{1}{2}\beta(t)\mathbf{X}_t$ and $g(t) = \sqrt{\beta(t)}$, with $\beta(t)$ being a noise schedule that determines the rate of data corruption. As $t \to T$, the distribution $p_t(\mathbf{X}_t)$ converges to a simple prior distribution (typically standard Gaussian), regardless of the initial data distribution $p_0(\mathbf{X}_0)$. The reverse process aims to invert this corruption using the reverse-time SDE:

$$\mathrm{d}\mathbf{X}_t = \left[\mathbf{f}(\mathbf{X}_t, t) - g^2(t)\nabla_{\mathbf{X}_t} \log p_t(\mathbf{X}_t)\right]\mathrm{d}\bar{t} + g(t)\mathrm{d}\bar{\mathbf{w}}_t, \tag{2}$$

where $\nabla_{\mathbf{X}_t} \log p_t(\mathbf{X}_t)$ denotes score function, $\mathrm{d}\bar{t}$ is an infinitesimal negative time step, and $\bar{\mathbf{w}}_t$ is the reverse-time Brownian motion (Freedman, 2012). The score function is approximated by a neural network trained via score matching (Kane et al., 2020), which minimizes the Fisher divergence between the model and the true data distribution to ensure accurate and data-consistent generation.

## 4 METHODOLOGY

In this section, we present the **DiffGDA** framework (Figure 1). We first introduce §4.1 *Domain-Guided Graph Diffusion*, which formulates adaptation as a continuous-time evolution. We then de-

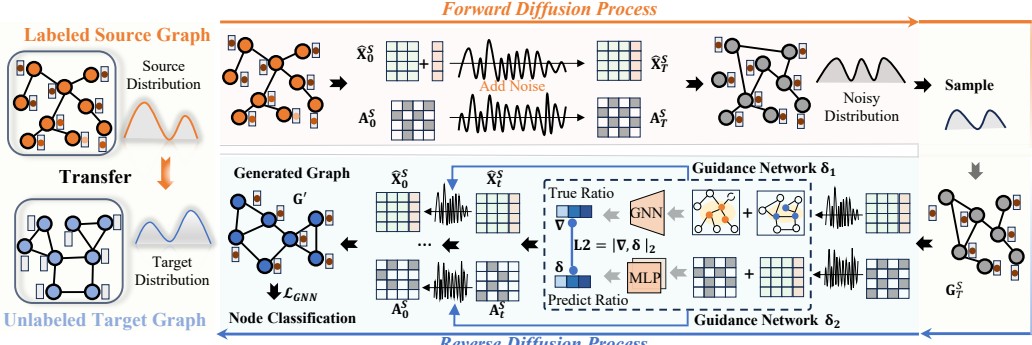

Figure 1: Overall architecture of our proposed DiffGDA framework: (1) In the forward diffusion process, labeled source graphs $\mathbf{G}^{\mathcal{S}}$ are progressively perturbed with noise, approximating a Gaussian distribution. (2) During the reverse diffusion process, a guidance network aligns predicted and true density ratios, reconstructing intermediate graphs to match the target distribution. The generated labeled graphs are then utilized for node classification training with a GNN.

scribe §4.2 *Practical Implementation*, covering the design of score and guidance networks. Finally, we detail the model optimization that integrates diffusion with GNN for end-to-end training.

## 4.1 DOMAIN-GUIDED GRAPH DIFFUSION PROCESS

**Forward Diffusion Process.** DiffGDA formulates GDA as a continuous-time stochastic process, governed by a system of SDEs. For a source graph denoted as $\mathbf{G}^{\mathcal{S}} = (\mathbf{X}^{\mathcal{S}}, \mathbf{A}^{\mathcal{S}}, \mathbf{Y}^{\mathcal{S}})$, we design a forward diffusion process that systematically perturbs both node features and adjacency matrices toward a tractable noise distribution. To integrate the source domain label knowledge into the diffusion process, we propose a structured feature concatenation to explicitly establish the semantic association between node features and labels. Specifically, the original node feature matrix $\mathbf{X}^{\mathcal{S}}$ and the label matrix $\mathbf{Y}^{\mathcal{S}}$ are concatenated along the channel dimension to construct augmented features: $\tilde{\mathbf{X}}^{\mathcal{S}} = [\mathbf{X}^{\mathcal{S}} || \mathbf{Y}^{\mathcal{S}}] \in \mathbb{R}^{N_{\mathcal{S}} \times (F+C)}$, where $||$ denotes the concatenation operation.

Formally, this process is defined as a continuous-time stochastic trajectory $\{\mathbf{G}_t^{\mathcal{S}} = (\tilde{\mathbf{X}}_t^{\mathcal{S}}, \mathbf{A}_t^{\mathcal{S}})\}_{t \in [0, \mathsf{T}]}$ over a fixed temporal domain $[0, \mathsf{T}]$, where $\mathbf{G}_0^{\mathcal{S}}$ represents the initial source graph. The complete dynamics of the forward process are governed by the following SDE:

$$\mathrm{d}\mathbf{G}_t^{\mathcal{S}} = \mathbf{f}_t(\mathbf{G}_t^{\mathcal{S}})\mathrm{d}t + g_t(\mathbf{G}_t^{\mathcal{S}})\mathrm{d}\mathbf{w}, \tag{3}$$

where $\mathbf{f}_t$ governs the deterministic drift, $g_t$ controls the noise injection rate. The Wiener process $\mathbf{w}$ drives the stochastic component. Overall, this process transforms the initial graph structure $\mathbf{G}_0^{\mathcal{S}} = (\tilde{\mathbf{X}}_0^{\mathcal{S}}, \mathbf{A}_0^{\mathcal{S}})$ into a simple noise distribution through designed drift term $\mathbf{f}_t$ and noise term $g_t \mathrm{d}\mathbf{w}$.

**Reverse Diffusion Process.** The reverse diffusion process aims to generate graphs that adhere to the target data distribution. It begins by sampling from the prior distribution and traversing the diffusion process of Eq. (3) backward in time. This reverse process is formulated as:

$$\mathrm{d}\mathbf{G}_t^{\mathcal{S}} = \left[\mathbf{f}_t(\mathbf{G}_t^{\mathcal{S}}) - g_t^2 \nabla_{\mathbf{G}_t^{\mathcal{S}}} \log p_t(\mathbf{G}_t^{\mathcal{S}})\right] \mathrm{d}\bar{t} + g_t \mathrm{d}\bar{\mathbf{w}}, \tag{4}$$

where $p_t$ denotes the marginal distribution under the forward diffusion process at time $t$, $\bar{\mathbf{w}}$ is a reverse-time standard Wiener process, and $\mathrm{d}\bar{t}$ represents an infinitesimal negative time step.

The key in reverse process is estimating the score function $\nabla_{\mathbf{G}_t^{\mathcal{S}}} \log p_t(\mathbf{G}_t^{\mathcal{S}})$. Typically, we can train a neural network $\mathbb{P}(\ell)$ via score matching to approximate this score function (Ouyang et al., 2024), thereby facilitating discrete-time generation via Eq. (4). However, this unconditional generation process only reconstructs the source graph distribution. In our scenario, the objective is to generate graphs conforming to a predefined target graph distribution $\mathbf{G}^{\mathcal{T}}$, which differs from the source distribution $\mathbf{G}^{\mathcal{S}}$. To address this issue, we propose an additional auxiliary guidance network $\mathbb{Q}(\boldsymbol{\delta})$, which steers generation process from the source distribution toward target distribution. Therefore,

the optimization objective in reverse process is expressed as:

$$\mathcal{L}_{\text{SDE}} = \mathcal{F}\left(\mathbb{P}(\boldsymbol{\ell}), \nabla_{\mathbf{G}_t^{\mathcal{S}}} \log p_t\left(\mathbf{G}_t^{\mathcal{S}}\right)\right) + \mathcal{F}\left(\mathbb{Q}(\boldsymbol{\delta}), \nabla_{\mathbf{G}_t^{\mathcal{S}}} \Psi\left(\mathbf{G}_t^{\mathcal{S}} \to \mathbf{G}_t^{\mathcal{T}}\right)\right), \tag{5}$$

where $\mathcal{F}(\cdot, \cdot)$ denotes a generic supervised loss (i.e., mean squared error), with the first argument as the trainable predictor and the second argument as its supervision signal. Crucially, $\Psi(\mathbf{G}_t^{\mathcal{S}} \to \mathbf{G}t^{\mathcal{T}})$ acts as a guidance potential, whose gradient $\nabla_{\mathbf{G}_t^{\mathcal{S}}} \Psi$ explicitly encourages alignment with the target domain. Building on the insights from Theorem 1, we can deduce the specific $\Psi\left(\mathbf{G}_t^{\mathcal{S}} \to \mathbf{G}_t^{\mathcal{T}}\right)$:

**Theorem 1.** *Let $\mathbf{G}^{\mathcal{S}}$ and $\mathbf{G}^{\mathcal{T}}$ denote the source and target graphs, respectively. Suppose the data distributions $p$ and $q$ define the forward diffusion processes on these two domains, respectively. Following (Ouyang et al., 2024), the optimal diffusion network $\mathbb{P}(\boldsymbol{\ell}^{\star})$ for the target graph $\mathbf{G}^{\mathcal{T}}$ satisfies:*

$$\mathbb{P}(\boldsymbol{\ell}^{\star}) = \nabla_{\mathbf{G}_t^{\mathcal{S}}} \log p_t(\mathbf{G}_t^{\mathcal{S}}) + \nabla_{\mathbf{G}_t^{\mathcal{S}}} \log \mathbb{E}_{p(\mathbf{G}_0^{\mathcal{S}}|\mathbf{G}_t^{\mathcal{S}})} q(\mathbf{G}_0^{\mathcal{T}})/p(\mathbf{G}_0^{\mathcal{S}}), \tag{6}$$

*where the first term is the score function of the source graph, and the second term represents the gradient of the density ratio between the target and source graph distributions. The density ratio is defined as the likelihood ratio between the target and source domains in the latent space.*

***Proof:*** *For a detailed derivation, please refer to the Appendix A.1.*

Consequently, the optimization objective in Eq. (5) can be rewritten as:

$$\mathcal{L}_{\text{SDE}} = \mathcal{F}\left(\mathbb{P}(\boldsymbol{\ell}), \nabla_{\mathbf{G}_t^{\mathcal{S}}} \log p_t\left(\mathbf{G}_t^{\mathcal{S}}\right)\right) + \mathcal{F}\left(\mathbb{Q}(\boldsymbol{\delta}), \nabla_{\mathbf{G}_t^{\mathcal{S}}} \log \mathbb{E}_{p(\mathbf{G}_0^{\mathcal{S}}|\mathbf{G}_t^{\mathcal{S}})} q(\mathbf{G}_0^{\mathcal{T}})/p(\mathbf{G}_0^{\mathcal{S}})\right). \tag{7}$$

The theorem demonstrates that our guidance network adaptively learns the optimal transformation path, directing the generative trajectory toward the target domain. This adaptive guidance enables the model to overcome the limitations of traditional discrete alignment methods by progressively steering the evolution toward the target domain, resulting in more coherent graph adaptation.

## 4.2 PRACTICAL IMPLEMENTATION

**Optimizing Score Network $\mathbb{P}(\boldsymbol{\ell})$.** To fit the score network, we first need to estimate the score function $\nabla_{\mathbf{G}_t^{\mathcal{S}}} \log p_t(\mathbf{G}_t^{\mathcal{S}}) \in \mathbb{R}^{N_{\mathcal{S}} \times (F+C)} \times \mathbb{R}^{N_{\mathcal{S}} \times N_{\mathcal{S}}}$, where $N_{\mathcal{S}}$ represents the node number in source graph, $F$ denotes the feature dimensionality, and $C$ corresponds to the class number. However, directly estimating this score function is computationally prohibitive due to the high dimensionality of the problem. Thus, we adopt the reverse-time diffusion process (Jo et al., 2022), which is mathematically equivalent to the formulation in Eq. (4) and is modeled by the following SDEs:

$$\begin{aligned} \mathrm{d}\tilde{\mathbf{X}}_t^{\mathcal{S}} &= \left[\mathbf{f}_{1,t}(\tilde{\mathbf{X}}_t^{\mathcal{S}}) - g_{1,t}^2 \nabla_{\tilde{\mathbf{X}}_t^{\mathcal{S}}} \log p_t(\tilde{\mathbf{X}}_t^{\mathcal{S}}, \mathbf{A}_t^{\mathcal{S}})\right] \mathrm{d}\bar{t} + g_{1,t}\, \mathrm{d}\bar{\mathbf{w}}_1, \\ \mathrm{d}\mathbf{A}_t^{\mathcal{S}} &= \left[\mathbf{f}_{2,t}(\mathbf{A}_t^{\mathcal{S}}) - g_{2,t}^2 \nabla_{\mathbf{A}_t^{\mathcal{S}}} \log p_t(\tilde{\mathbf{X}}_t^{\mathcal{S}}, \mathbf{A}_t^{\mathcal{S}})\right] \mathrm{d}\bar{t} + g_{2,t}\, \mathrm{d}\bar{\mathbf{w}}_2, \end{aligned} \tag{8}$$

where $\mathbf{f}_{1,t}$ and $\mathbf{f}_{2,t}$ are linear drift coefficients satisfying $\mathbf{f}_t(\tilde{\mathbf{X}}, \mathbf{A}) = (\mathbf{f}_{1,t}(\tilde{\mathbf{X}}), \mathbf{f}_{2,t}(\mathbf{A}))$, $g_{1,t}$ and $g_{2,t}$ are scalar diffusion coefficients, and $\bar{\mathbf{w}}_1, \bar{\mathbf{w}}_2$ represent reverse-time standard Wiener processes. This formulation effectively decomposes the original high-dimensional score function into two partial score functions: $\nabla_{\tilde{\mathbf{X}}_t^{\mathcal{S}}} \log p_t(\tilde{\mathbf{X}}_t^{\mathcal{S}}, \mathbf{A}_t^{\mathcal{S}})$ and $\nabla_{\mathbf{A}_t^{\mathcal{S}}} \log p_t(\tilde{\mathbf{X}}_t^{\mathcal{S}}, \mathbf{A}_t^{\mathcal{S}})$. By splitting the computation, the problem becomes more tractable, as it reduces the complexity of estimating the high-dimensional score function into manageable components, which can be estimated using the Variance Exploding (VE) (Yang et al., 2023) technique to compute these two gradients. Consequently, the score network $\mathbb{P}(\boldsymbol{\ell})$ is also divided into two parts: $\mathbb{P}(\boldsymbol{\ell}) \triangleq (\mathbb{P}(\boldsymbol{\ell}_1), \mathbb{P}(\boldsymbol{\ell}_2))$, where $\mathbb{P}(\boldsymbol{\ell}1)$ is responsible for computing the score function associated with the node features, and $\mathbb{P}(\boldsymbol{\ell}_2)$ is tasked with computing the score function for the graph adjacency structure, "$\triangleq$" denotes a decomposition. To model these two score functions, we utilize two independent multilayer perceptron (MLP) networks:

$$\mathbb{P}(\boldsymbol{\ell}_1) = \text{MLP}\left(\left[\{\mathbf{H}_i\}_{i=0}^{L}\right]\right), \quad \mathbb{P}(\boldsymbol{\ell}_2) = \text{MLP}\left(\left[\{\text{GMH}(\mathbf{H}_i, \mathbf{A}_t^{\kappa})\}_{i=0,\kappa=1}^{L,K}\right]\right), \tag{9}$$

where $\mathbf{A}_t^{\kappa}$ denotes the higher-order adjacency matrices, and $\mathbf{H}_{i+1} = \text{GNN}(\mathbf{H}_i, \mathbf{A}_t)$ with $\mathbf{H}_0 = \tilde{\mathbf{X}}_0$ as the input. The GMH refers to the graph multi-head attention block, $K$ is the number of GMH

layers, and $L$ is the number of GNN layers. To approximate these score functions using score network $\mathbb{P}(\boldsymbol{\ell})$, we define the following training objectives for nodes and edges, respectively:

$$
\begin{aligned}
\boldsymbol{\ell}_1^\star &= \arg\min_{\boldsymbol{\ell}_1} \mathbb{E}_t \left\{ \lambda_{1,t} \mathbb{E}_{\mathbf{G}_0^{\mathcal{S}}} \mathbb{E}_{\mathbf{G}_t^{\mathcal{S}}|\mathbf{G}_0^{\mathcal{S}}} \left\| \mathbb{P}(\boldsymbol{\ell}_1) - \nabla_{\mathbf{X}_t^{\mathcal{S}}} \log p_t(\tilde{\mathbf{X}}_t^{\mathcal{S}}|\tilde{\mathbf{X}}_0^{\mathcal{S}}) \right\|_2^2 \right\}, \\
\boldsymbol{\ell}_2^\star &= \arg\min_{\boldsymbol{\ell}_2} \mathbb{E}_t \left\{ \lambda_{2,t} \mathbb{E}_{\mathbf{G}_0^{\mathcal{S}}} \mathbb{E}_{\mathbf{G}_t^{\mathcal{S}}|\mathbf{G}_0^{\mathcal{S}}} \left\| \mathbb{P}(\boldsymbol{\ell}_2) - \nabla_{\mathbf{A}_t^{\mathcal{S}}} \log p_t(\mathbf{A}_t|\mathbf{A}_0^{\mathcal{S}}) \right\|_2^2 \right\},
\end{aligned}
\tag{10}
$$

where $\lambda_{1,t}$ and $\lambda_{2,t}$ are weighting functions and $t$ is uniformly sampled from $[0, \mathsf{T}]$, the $\mathbb{E}$ can be efficiently computed using the Monte Carlo estimate (McCool, 1999) with the samples $(t, \mathbf{G}_0, \mathbf{G}_t)$.

**Optimizing Guidance Network $\mathbb{Q}(\boldsymbol{\delta})$.** We begin by formulating the gradient estimation objective $\nabla_{\mathbf{G}_t^{\mathcal{S}}} \log \mathbb{E}_{p(\mathbf{G}_0^{\mathcal{S}}|\mathbf{G}_t^{\mathcal{S}})} q(\mathbf{G}_0^{\mathcal{T}})/p(\mathbf{G}_0^{\mathcal{S}})$. This objective can be decomposed into two key components: (1) the density ratio $q(\mathbf{G}_0^{\mathcal{T}})/p(\mathbf{G}_0^{\mathcal{S}})$ between target and source domains, and (2) the expectation operator over the posterior distribution $\mathbb{E}_{p(\mathbf{G}_0^{\mathcal{S}}|\mathbf{G}_t^{\mathcal{S}})}$. To address the intractability of direct computation, we estimate these two items respectively through the following methods inspired by (Ouyang et al., 2024): (1) We approximate the density ratio through domain discrimination. A graph neural classifier $\mathcal{C}_{\text{gnn}}: \mathbf{G}^{\mathcal{S}} \cup \mathbf{G}^{\mathcal{T}} \to \mathbf{y}$ is trained to distinguish between source and target domain nodes, using a subset of nodes from each domain. Let $\mathbf{y}(\mathbf{x}) \in [0, 1]$ denote the classifier's output indicating source membership. The density ratio can then be estimated as: $q(\mathbf{G}_0^{\mathcal{T}})/p(\mathbf{G}_0^{\mathcal{S}}) \approx (1 - \mathbf{y}(\mathbf{x}))/\mathbf{y}(\mathbf{x})$. (2) We compute the term $\mathbb{E}_{p(\mathbf{G}_0^{\mathcal{S}}|\mathbf{G}_t^{\mathcal{S}})} q(\mathbf{G}_0^{\mathcal{T}})/p(\mathbf{G}_0^{\mathcal{S}})$ using Monte Carlo sampling. This process involves generating multiple perturbed graph instances from the joint distribution $p(\mathbf{G}_0^{\mathcal{S}}, \mathbf{G}_t^{\mathcal{S}})$ and then averaging the computed gradients across these samples (Lu et al., 2023). We approximate the true gradient by sampling from this distribution. The sampling details are provided in Appendix B.

Similarly to the score network $\mathbb{P}(\boldsymbol{\ell})$, we approximate the guidance function using a parameterized network $\mathbb{Q}(\boldsymbol{\delta})$, where $\mathbb{Q}(\boldsymbol{\delta}) \triangleq (\mathbb{Q}(\boldsymbol{\delta}_1), \mathbb{Q}(\boldsymbol{\delta}_2))$. Here, $\mathbb{Q}(\boldsymbol{\delta}_1)$ acts as the domain estimator for node-level features, while $\mathbb{Q}(\boldsymbol{\delta}_2)$ serves as the domain estimator for the adjacency matrix. The two components are formally expressed as follows:

$$
\mathbb{Q}(\boldsymbol{\delta}_1) = \text{MLP}\left(\tilde{\mathbf{X}}_t^{\mathcal{S}}, t\right), \quad \mathbb{Q}(\boldsymbol{\delta}_2) = \text{MLP}\left(\mathbf{A}_t^{\mathcal{S}}, t\right).
\tag{11}
$$

Thus, the final learning objective for the guidance network $\mathbb{Q}(\boldsymbol{\delta})$ can be described as follows:

$$
\begin{aligned}
\boldsymbol{\delta}_1^\star &= \arg\min_{\boldsymbol{\delta}_1} \mathbb{E}_{p(\mathbf{G}_0^{\mathcal{S}}, \mathbf{G}_t^{\mathcal{S}})} \left\| \mathbb{Q}(\boldsymbol{\delta}_1) - q(\mathbf{X}_0^{\mathcal{T}})/p(\mathbf{X}_0^{\mathcal{S}}) \right\|_2^2, \\
\boldsymbol{\delta}_2^\star &= \arg\min_{\boldsymbol{\delta}_2} \mathbb{E}_{p(\mathbf{G}_0^{\mathcal{S}}, \mathbf{G}_t^{\mathcal{S}})} \left\| \mathbb{Q}(\boldsymbol{\delta}_2) - q(\mathbf{A}_0^{\mathcal{T}})/p(\mathbf{A}_0^{\mathcal{S}}) \right\|_2^2.
\end{aligned}
\tag{12}
$$

In this manner, we can achieve an accurate estimation of the guiding function in Eq. (7) as:

$$
\mathbb{Q}(\boldsymbol{\delta}^\star) = (\mathbb{Q}(\boldsymbol{\delta}_1^\star), \mathbb{Q}(\boldsymbol{\delta}_2^\star)) = \log \mathbb{E}_{p(\mathbf{G}_0^{\mathcal{S}}|\mathbf{G}_t^{\mathcal{S}})} q(\mathbf{G}_0^{\mathcal{T}})/p(\mathbf{G}_0^{\mathcal{S}}).
\tag{13}
$$

The detailed proof can be found in Appendix A.2.

Applying diffusion indiscriminately to all nodes is computationally costly and may weaken informative signals. To mitigate this, we adopt a **targeted stochastic diffusion** strategy that selectively perturbs a subset of nodes while keeping the rest intact, i.e., $p(\mathbf{G}_0^{\mathcal{T}}) \sim p(\mathbf{G}^{\mathcal{T}})$. A hyperparameter $\alpha$ controls the diffusion ratio, balancing efficiency and information preservation.

**Training on Generated Graph $\mathbf{G}'$.** After training the diffusion model, we obtain $\mathbf{G}' = (\mathbf{X}', \mathbf{A}', \mathbf{Y}')$ with node features, adjacency matrix, and labels. Since $\mathbf{G}'$ may still differ from the target graph, we adopt an additional MMD alignment (Huang et al., 2024). The training loss is:

$$
\mathcal{L}_{\text{GNN}} = \mathcal{L}_{\text{CE}}(\text{GNN}(\mathbf{X}', \mathbf{A}'), \mathbf{Y}') + \eta\, \mathcal{L}_{\text{MMD}}\left(\text{GNN}(\mathbf{X}', \mathbf{A}'), \text{GNN}(\mathbf{X}^{\mathcal{T}}, \mathbf{A}^{\mathcal{T}})\right),
\tag{14}
$$

where $\mathcal{L}_{\text{CE}}$ is the cross-entropy loss, $\mathcal{L}_{\text{MMD}}$ aligns $\mathbf{G}'$ with $\mathbf{G}^{\mathcal{T}}$, and $\eta$ is a weighting factor.

The design of $\mathbf{G}'$ generation offers two main advantages in GDA. First, it allows the diffusion model to learn class-aware dynamics directly in the joint feature-label space, rather than aligning feature-level distributions alone, which can be ambiguous without labels. Second, by reconstructing labels with features, the model can better capture the relationship between node attributes, structure, and class membership, leading to improved node classification performance on the target domain.

| Methods | ACMv9 (A), Citationv1 (C), DBLPv7 (D) | | | | | | | | | | | | Avg. |
|---|---|---|---|---|---|---|---|---|---|---|---|---|---|
| | $A \rightarrow C$ | | $A \rightarrow D$ | | $C \rightarrow A$ | | $C \rightarrow D$ | | $D \rightarrow A$ | | $D \rightarrow C$ | | |
| | Mi-F1 | Ma-F1 | Mi-F1 | Ma-F1 | Mi-F1 | Ma-F1 | Mi-F1 | Ma-F1 | Mi-F1 | Ma-F1 | Mi-F1 | Ma-F1 | |
| GAT (ICLR'18) | $62.77_{\pm2.24}$ | $59.60_{\pm2.82}$ | $60.29_{\pm1.33}$ | $55.09_{\pm1.16}$ | $58.35_{\pm1.51}$ | $58.06_{\pm1.88}$ | $67.07_{\pm2.05}$ | $63.82_{\pm1.85}$ | $54.33_{\pm2.17}$ | $52.99_{\pm2.36}$ | $63.24_{\pm2.21}$ | $59.96_{\pm2.86}$ | 59.63 |
| GIN (ICLR'19) | $69.95_{\pm0.90}$ | $62.34_{\pm1.08}$ | $64.69_{\pm1.43}$ | $53.08_{\pm2.06}$ | $62.63_{\pm0.23}$ | $61.34_{\pm0.62}$ | $68.54_{\pm0.31}$ | $64.83_{\pm0.44}$ | $58.18_{\pm0.60}$ | $51.36_{\pm2.36}$ | $69.91_{\pm1.83}$ | $63.09_{\pm1.92}$ | 62.50 |
| GCN (ICLR'17) | $70.82_{\pm1.26}$ | $66.49_{\pm2.21}$ | $65.05_{\pm2.15}$ | $59.53_{\pm0.44}$ | $65.44_{\pm1.14}$ | $65.06_{\pm1.38}$ | $69.46_{\pm0.83}$ | $65.80_{\pm1.82}$ | $59.92_{\pm0.72}$ | $58.95_{\pm0.57}$ | $66.83_{\pm0.94}$ | $64.66_{\pm1.01}$ | 64.83 |
| DANE (IJCAI'19) | $69.77_{\pm2.14}$ | $66.89_{\pm2.90}$ | $62.41_{\pm2.15}$ | $59.09_{\pm2.60}$ | $63.93_{\pm1.32}$ | $64.04_{\pm2.01}$ | $65.05_{\pm2.07}$ | $60.34_{\pm3.01}$ | $58.21_{\pm1.17}$ | $57.98_{\pm1.47}$ | $67.41_{\pm2.29}$ | $62.37_{\pm2.47}$ | 63.12 |
| UDAGCN (WWW'20) | $80.68_{\pm0.31}$ | $78.74_{\pm0.55}$ | $74.66_{\pm0.93}$ | $\underline{72.59}_{\pm1.05}$ | $73.46_{\pm0.40}$ | $74.37_{\pm0.35}$ | $76.97_{\pm0.31}$ | $\underline{75.56}_{\pm0.42}$ | $69.36_{\pm0.31}$ | $70.10_{\pm0.60}$ | $77.81_{\pm0.50}$ | $76.09_{\pm0.78}$ | 75.03 |
| AdaGCN (TKDE'22) | $68.07_{\pm0.86}$ | $64.15_{\pm1.88}$ | $66.72_{\pm1.07}$ | $61.97_{\pm1.25}$ | $63.25_{\pm1.15}$ | $63.22_{\pm1.25}$ | $70.97_{\pm0.48}$ | $66.43_{\pm0.43}$ | $60.68_{\pm0.67}$ | $58.99_{\pm1.42}$ | $65.83_{\pm2.33}$ | $58.27_{\pm1.49}$ | 64.05 |
| StruRW (ICML'23) | $60.48_{\pm1.41}$ | $54.49_{\pm1.41}$ | $59.63_{\pm0.34}$ | $52.54_{\pm0.57}$ | $56.18_{\pm0.86}$ | $51.77_{\pm2.01}$ | $62.50_{\pm1.40}$ | $56.74_{\pm2.01}$ | $52.25_{\pm1.10}$ | $44.68_{\pm1.25}$ | $57.55_{\pm0.84}$ | $50.47_{\pm1.25}$ | 54.94 |
| GRADE (AAAI'23) | $75.02_{\pm0.48}$ | $71.66_{\pm0.50}$ | $68.17_{\pm0.25}$ | $63.05_{\pm0.41}$ | $68.96_{\pm0.10}$ | $68.43_{\pm0.14}$ | $73.48_{\pm0.30}$ | $69.76_{\pm0.92}$ | $61.72_{\pm0.36}$ | $56.45_{\pm0.60}$ | $71.69_{\pm0.90}$ | $66.54_{\pm0.75}$ | 67.91 |
| PairAlign (ICML'24) | $60.25_{\pm0.89}$ | $55.11_{\pm1.02}$ | $59.58_{\pm0.58}$ | $53.80_{\pm0.85}$ | $56.02_{\pm0.85}$ | $51.57_{\pm1.36}$ | $63.49_{\pm0.70}$ | $59.02_{\pm0.82}$ | $51.83_{\pm0.69}$ | $46.47_{\pm1.13}$ | $58.60_{\pm0.42}$ | $53.66_{\pm0.47}$ | 55.78 |
| GraphAlign (KDD'24) | $75.18_{\pm0.62}$ | $71.09_{\pm1.21}$ | $68.81_{\pm0.78}$ | $65.51_{\pm1.21}$ | $65.21_{\pm0.55}$ | $61.37_{\pm0.20}$ | $72.21_{\pm0.45}$ | $71.18_{\pm1.32}$ | $61.66_{\pm1.23}$ | $62.33_{\pm0.45}$ | $68.85_{\pm0.56}$ | $65.56_{\pm0.21}$ | 67.41 |
| A2GNN (AAAI'24) | $\underline{80.93}_{\pm0.52}$ | $78.06_{\pm0.73}$ | $\underline{75.94}_{\pm0.33}$ | $71.31_{\pm0.34}$ | $\underline{75.09}_{\pm0.43}$ | $76.41_{\pm0.54}$ | $77.16_{\pm0.23}$ | $73.28_{\pm0.32}$ | $\underline{73.21}_{\pm0.44}$ | $74.48_{\pm0.06}$ | $\underline{79.72}_{\pm0.63}$ | $76.01_{\pm0.79}$ | $\underline{75.97}$ |
| GGDA (Arxiv'25) | $79.33_{\pm1.00}$ | $76.02_{\pm1.93}$ | $73.58_{\pm1.73}$ | $68.97_{\pm3.80}$ | $73.50_{\pm0.60}$ | $73.91_{\pm0.92}$ | $76.62_{\pm0.59}$ | $72.94_{\pm0.64}$ | $70.94_{\pm0.11}$ | $72.27_{\pm0.32}$ | $77.95_{\pm0.54}$ | $75.24_{\pm1.52}$ | 74.27 |
| TDSS (AAAI'25) | $80.41_{\pm0.71}$ | $\underline{79.10}_{\pm1.64}$ | $74.04_{\pm3.64}$ | $71.59_{\pm3.08}$ | $72.88_{\pm2.83}$ | $76.09_{\pm1.06}$ | $\underline{77.23}_{\pm2.71}$ | $74.96_{\pm0.75}$ | $72.38_{\pm1.95}$ | $73.23_{\pm3.09}$ | $79.04_{\pm3.61}$ | $\underline{77.70}_{\pm2.68}$ | 75.72 |
| DGSDA (ICML'25) | $78.59_{\pm0.55}$ | $77.03_{\pm0.52}$ | $73.71_{\pm0.24}$ | $72.63_{\pm0.25}$ | $73.96_{\pm0.70}$ | $71.61_{\pm0.80}$ | $76.56_{\pm0.25}$ | $75.52_{\pm0.34}$ | $72.69_{\pm0.34}$ | $71.82_{\pm0.45}$ | $77.41_{\pm0.11}$ | $76.13_{\pm0.23}$ | 74.75 |
| GAA (ICLR'25) | $80.03_{\pm0.37}$ | $75.85_{\pm0.65}$ | $73.32_{\pm0.67}$ | $68.07_{\pm1.09}$ | $73.15_{\pm0.32}$ | $73.57_{\pm0.55}$ | $76.04_{\pm0.39}$ | $71.40_{\pm0.16}$ | $68.32_{\pm0.43}$ | $62.66_{\pm0.93}$ | $78.27_{\pm0.31}$ | $72.74_{\pm0.76}$ | 72.65 |
| **DiffGDA (Ours)** | $\mathbf{82.28}_{\pm0.46}$ | $\mathbf{81.07}_{\pm0.37}$ | $\mathbf{76.70}_{\pm0.92}$ | $\mathbf{73.20}_{\pm1.68}$ | $\mathbf{75.75}_{\pm0.30}$ | $\mathbf{77.04}_{\pm0.47}$ | $\mathbf{78.11}_{\pm0.40}$ | $\mathbf{77.09}_{\pm1.09}$ | $\mathbf{74.55}_{\pm0.20}$ | $\mathbf{75.96}_{\pm0.06}$ | $\mathbf{80.71}_{\pm0.20}$ | $\mathbf{78.53}_{\pm0.58}$ | **77.58** |

| Methods | USA (U), Brazil (B), Europe (E) | | | | | | | | | | | | Avg. |
|---|---|---|---|---|---|---|---|---|---|---|---|---|---|
| | $U \rightarrow B$ | | $U \rightarrow E$ | | $B \rightarrow U$ | | $B \rightarrow E$ | | $E \rightarrow U$ | | $E \rightarrow B$ | | |
| | Mi-F1 | Ma-F1 | Mi-F1 | Ma-F1 | Mi-F1 | Ma-F1 | Mi-F1 | Ma-F1 | Mi-F1 | Ma-F1 | Mi-F1 | Ma-F1 | |
| GAT (ICLR'18) | $50.53_{\pm3.25}$ | $48.01_{\pm1.03}$ | $40.25_{\pm3.82}$ | $35.57_{\pm3.78}$ | $44.25_{\pm1.30}$ | $41.25_{\pm1.57}$ | $46.67_{\pm1.99}$ | $46.18_{\pm1.23}$ | $43.90_{\pm1.69}$ | $42.13_{\pm1.72}$ | $50.84_{\pm3.22}$ | $49.02_{\pm4.50}$ | 44.88 |
| GIN (ICLR'19) | $33.13_{\pm2.96}$ | $23.29_{\pm4.55}$ | $32.28_{\pm4.55}$ | $22.50_{\pm4.12}$ | $35.87_{\pm1.96}$ | $28.06_{\pm1.93}$ | $32.98_{\pm2.54}$ | $26.03_{\pm4.30}$ | $36.07_{\pm2.44}$ | $28.03_{\pm2.24}$ | $33.89_{\pm4.03}$ | $26.21_{\pm4.23}$ | 29.86 |
| GCN (ICLR'17) | $56.34_{\pm3.75}$ | $55.12_{\pm5.22}$ | $44.71_{\pm0.52}$ | $42.63_{\pm1.04}$ | $40.22_{\pm1.21}$ | $31.21_{\pm2.51}$ | $48.92_{\pm1.48}$ | $46.52_{\pm1.77}$ | $44.67_{\pm0.99}$ | $37.39_{\pm4.21}$ | $58.47_{\pm2.39}$ | $57.27_{\pm3.13}$ | 46.96 |
| DANE (IJCAI'19) | $44.12_{\pm1.56}$ | $41.23_{\pm1.63}$ | $39.90_{\pm3.25}$ | $34.24_{\pm4.76}$ | $42.20_{\pm4.43}$ | $35.73_{\pm5.31}$ | $38.45_{\pm3.90}$ | $33.57_{\pm3.90}$ | $36.13_{\pm7.26}$ | $28.26_{\pm7.44}$ | $46.56_{\pm0.48}$ | $39.94_{\pm0.80}$ | 38.36 |
| UDAGCN (WWW'20) | $58.17_{\pm2.07}$ | $56.82_{\pm2.59}$ | $44.51_{\pm0.41}$ | $41.85_{\pm0.86}$ | $42.17_{\pm1.14}$ | $34.65_{\pm1.26}$ | $47.97_{\pm1.72}$ | $47.60_{\pm1.75}$ | $42.79_{\pm1.47}$ | $41.83_{\pm1.07}$ | $62.29_{\pm3.79}$ | $61.97_{\pm4.15}$ | 48.55 |
| AdaGCN (TKDE'22) | $65.65_{\pm1.93}$ | $\underline{66.15}_{\pm1.38}$ | $50.63_{\pm0.55}$ | $\underline{51.45}_{\pm0.93}$ | $46.87_{\pm1.00}$ | $43.89_{\pm1.07}$ | $54.44_{\pm0.93}$ | $\underline{54.92}_{\pm0.91}$ | $48.62_{\pm0.53}$ | $44.37_{\pm0.64}$ | $\underline{73.74}_{\pm2.24}$ | $\underline{73.34}_{\pm2.46}$ | 56.17 |
| StruRW (ICML'23) | $62.44_{\pm1.96}$ | $62.25_{\pm2.61}$ | $39.70_{\pm1.71}$ | $37.26_{\pm2.15}$ | $41.45_{\pm1.30}$ | $40.02_{\pm1.56}$ | $38.30_{\pm1.52}$ | $36.84_{\pm1.54}$ | $41.21_{\pm1.50}$ | $38.10_{\pm0.78}$ | $55.11_{\pm0.75}$ | $52.15_{\pm0.57}$ | 45.40 |
| GRADE (AAAI'23) | $57.71_{\pm0.78}$ | $56.16_{\pm0.96}$ | $48.97_{\pm0.20}$ | $46.86_{\pm0.22}$ | $43.31_{\pm0.58}$ | $38.68_{\pm0.41}$ | $\underline{55.94}_{\pm0.19}$ | $53.93_{\pm0.33}$ | $46.64_{\pm0.80}$ | $42.47_{\pm0.65}$ | $63.05_{\pm0.37}$ | $60.13_{\pm0.36}$ | 51.15 |
| PairAlign (ICML'24) | $66.26_{\pm1.77}$ | $65.35_{\pm2.65}$ | $39.50_{\pm0.95}$ | $35.61_{\pm0.45}$ | $41.92_{\pm2.30}$ | $40.89_{\pm2.33}$ | $38.10_{\pm2.60}$ | $37.57_{\pm3.00}$ | $39.63_{\pm0.94}$ | $37.71_{\pm0.48}$ | $55.57_{\pm0.75}$ | $53.26_{\pm1.27}$ | 45.95 |
| GraphAlign (KDD'24) | $62.54_{\pm0.80}$ | $61.33_{\pm1.21}$ | $\underline{52.18}_{\pm0.17}$ | $50.09_{\pm2.33}$ | $\underline{50.33}_{\pm0.77}$ | $\underline{47.11}_{\pm1.17}$ | $55.23_{\pm2.33}$ | $53.19_{\pm0.78}$ | $\underline{54.35}_{\pm0.33}$ | $\underline{49.31}_{\pm0.37}$ | $71.02_{\pm0.48}$ | $69.97_{\pm1.22}$ | $\underline{56.39}$ |
| A2GNN (AAAI'24) | $64.24_{\pm1.37}$ | $61.08_{\pm2.57}$ | $46.62_{\pm1.61}$ | $43.51_{\pm3.94}$ | $47.66_{\pm5.51}$ | $38.60_{\pm7.73}$ | $47.67_{\pm1.35}$ | $42.80_{\pm3.55}$ | $52.72_{\pm1.16}$ | $48.39_{\pm1.07}$ | $54.81_{\pm1.63}$ | $46.76_{\pm2.47}$ | 49.57 |
| GGDA (Arxiv'25) | $62.44_{\pm2.36}$ | $60.05_{\pm1.19}$ | $48.17_{\pm1.72}$ | $43.48_{\pm3.08}$ | $47.03_{\pm1.29}$ | $38.80_{\pm0.74}$ | $48.67_{\pm0.92}$ | $38.88_{\pm2.98}$ | $49.94_{\pm1.52}$ | $47.09_{\pm2.03}$ | $65.80_{\pm1.70}$ | $62.46_{\pm3.17}$ | 51.07 |
| TDSS (AAAI'25) | $\underline{67.43}_{\pm1.62}$ | $63.36_{\pm1.33}$ | $52.05_{\pm1.01}$ | $48.17_{\pm1.50}$ | $47.62_{\pm0.77}$ | $36.46_{\pm2.63}$ | $51.80_{\pm0.12}$ | $40.38_{\pm1.15}$ | $46.08_{\pm0.51}$ | $36.55_{\pm2.14}$ | $55.73_{\pm1.08}$ | $45.27_{\pm1.12}$ | 49.24 |
| DGSDA (ICML'25) | $61.22_{\pm1.63}$ | $60.28_{\pm1.55}$ | $49.35_{\pm2.76}$ | $48.20_{\pm1.46}$ | $54.91_{\pm0.51}$ | $53.76_{\pm0.78}$ | $52.56_{\pm1.64}$ | $42.58_{\pm1.73}$ | $49.76_{\pm0.82}$ | $47.59_{\pm0.76}$ | $60.15_{\pm0.57}$ | $59.30_{\pm0.63}$ | 52.28 |
| GAA (ICLR'25) | $64.89_{\pm2.36}$ | $59.28_{\pm2.67}$ | $51.70_{\pm1.97}$ | $50.16_{\pm1.01}$ | $52.50_{\pm1.67}$ | $50.18_{\pm1.03}$ | $48.22_{\pm1.07}$ | $42.76_{\pm0.64}$ | $48.70_{\pm2.34}$ | $47.17_{\pm1.30}$ | $56.07_{\pm1.51}$ | $55.34_{\pm2.73}$ | 51.98 |
| **DiffGDA (Ours)** | $\mathbf{71.76}_{\pm1.25}$ | $\mathbf{72.21}_{\pm0.98}$ | $\mathbf{54.18}_{\pm0.58}$ | $\mathbf{52.58}_{\pm1.00}$ | $\mathbf{54.37}_{\pm0.88}$ | $\mathbf{50.37}_{\pm1.27}$ | $\mathbf{57.15}_{\pm0.85}$ | $\mathbf{57.56}_{\pm0.12}$ | $\mathbf{56.20}_{\pm0.63}$ | $\mathbf{52.93}_{\pm0.34}$ | $\mathbf{74.81}_{\pm1.63}$ | $\mathbf{74.84}_{\pm0.85}$ | **60.75** |

Table 1: Node classification performance ($\% \pm \sigma$) on the Citation and Airport domains. The highest scores are highlighted in **bold**, while the second-highest scores are underlined.

**Model Optimization.** Overall, the objective consists of optimizing both GNN and diffusion model in an end-to-end manner. Specifically, the optimization problem can be formulated as:

$$\min_{\boldsymbol{\ell}_1, \boldsymbol{\ell}_2, \boldsymbol{\delta}_1, \boldsymbol{\delta}_2, \boldsymbol{\theta}} \mathcal{L}_{\text{SDE}} + \mathcal{L}_{\text{GNN}}, \tag{15}$$

where $\boldsymbol{\ell}_1, \boldsymbol{\ell}_2, \boldsymbol{\delta}_1, \boldsymbol{\delta}_2$ represent the diffusion parameters, and $\boldsymbol{\theta}$ denotes the parameters of the GNN.

**Complexity Analysis.** The overall computational complexity of DiffGDA is governed by two main components: the diffusion model and the graph neural network (GNN). For the diffusion model, which is implemented as a multilayer perceptron (MLP), the complexity is $\mathcal{O}(\mathsf{T} \cdot n^2)$, where $\mathsf{T}$ denotes the number of diffusion steps and $n$ is diffusion node number. Importantly, the targeted diffusion strategy controls $n = \alpha |\mathcal{V}^{\mathsf{S}}|$, making $\alpha$ the key knob to balance efficiency. The GNN introduces a complexity of $\mathcal{O}(L \cdot (|\mathcal{V}^{\mathcal{S}}| + |\mathcal{E}^{\mathcal{S}}|))$, where $L$ is the number of layers, $|\mathcal{V}^{\mathcal{S}}|$ is the number of source nodes, and $|\mathcal{E}^{\mathcal{S}}|$ is the number of edges in source graph. Thus, the total complexity is $\mathcal{O}(\mathsf{T} \cdot n^2 + L \cdot (|\mathcal{V}^{\mathcal{S}}| + |\mathcal{E}^{\mathcal{S}}|))$. The **error bound** analysis is provided in Appendix C.

**Remark.** (Discussion on Computational Cost) Firstly, despite being diffusion-based, our framework introduces a stochastic diffusion strategy that processes subsets of nodes while preserving critical source information. With an adjustable sampling ratio, it balances computational cost and performance, achieving faster convergence and higher accuracy compared to other methods. Secondly, the model complexity is $\mathcal{O}(\mathsf{T} \cdot n^2 + L \cdot (|\mathcal{V}^S| + |\mathcal{E}^S|))$, where $\mathsf{T}$ is the diffusion steps, $n$ the sampled nodes, and $L$ the GNN layers. In practice, $\mathsf{T} \leq 100$ and $n$ is controlled via stochastic sampling, keeping $n^2$ manageable. The shallow GNNs further enhance efficiency. Thus, DiffGDA achieves competitive performance with modest computational resources, making it practical and efficient.

| Domain | # Datasets | # Nodes | # Edges | # Feat Dims | # Labels |
|---|---|---|---|---|---|
| Citation | ACMv9 (A) | 9,360 | 31,112 | 6,775 | 5 |
| | Citationv1 (C) | 8,935 | 30,196 | | |
| | DBLPv7 (D) | 5,484 | 16,234 | | |
| Airport | USA (U) | 1,190 | 27,198 | 241 | 4 |
| | Brazil (B) | 131 | 2,148 | | |
| | Europe (E) | 399 | 11,990 | | |
| Social | Blog1 (B1) | 2,300 | 66,942 | 8,189 | 6 |
| | Blog2 (B2) | 2,896 | 107,672 | | |

Table 2: Statistics of the eight real-world datasets.

## 5 EXPERIMENTS

In this section, we empirically evaluate the DiffGDA framework through experiments addressing five key research questions: **RQ1**: How does DiffGDA compare to state-of-the-art methods? **RQ2**: Which part of the DiffGDA model majorly contributed to the effective prediction? **RQ3**: How do key parameters impact model performance? **RQ4**: What are the computational efficiency and resource overhead of DiffGDA? **RQ5**: How can we intuitively understand DiffGDA's strengths?

### 5.1 EXPERIMENTAL SETUP

**(1) Datasets.** We employ three domains, as detailed in Table 2, which include a total of eight commonly used datasets: Citation (ACMv9, Citationv1, DBLPv7) (Dai et al., 2022), Airport (USA, Brazil, Europe) (Ribeiro et al., 2017), and Social (Blog1, Blog2) (Liu et al., 2024a). **(2) Baselines.** To validate the effectiveness of our model, we compare against two categories of baseline methods: (1) source-only graph neural networks including GAT, GIN , and GCN, which are trained exclusively on the source domain and directly transferred to the target domain; and (2) graph domain adaptation methods: DANE, UDAGCN, AdaGCN, StruRW, GRADE, PairAlign, GraphAlign, A2GNN, GGDA, TDSS, DGSDA and GAA. The complete experimental setup is provided in Appendix D.

### 5.2 MAIN EXPERIMENTAL RESULTS (**RQ1**)

We evaluate the performance of DiffGDA framework on node classification tasks across three domains, as shown in Table 1 (with additional Social results in Appendix E.1). The results consistently demonstrate that DiffGDA outperforms state-of-the-art baselines across a variety of transfer scenarios. On the first three datasets, DiffGDA achieves the highest average score, showcasing its superiority in modeling complex cross-domain transitions via continuous-time diffusion processes. Unlike traditional discrete intermediate approaches (i.e., GGDA, GraphAlign), DiffGDA effectively captures nonlinear and multi-scale structural discrepancies. Likewise, in the USA, Brazil, and Europe transfer settings, DiffGDA maintains strong generalization capability, significantly outperforming competitive methods such as GraphAlign and TDSS. These findings validate the efficacy of our generative continuous evolution perspective, which reduces information distortion and enhances structural alignment in GDA. The significance tests ($p < 0.05$) are provided in Appendix E.2.

### 5.3 ABLATION STUDIES (**RQ2**)

To evaluate the contribution of each module in DiffGDA framework, we conduct ablation studies on four cross-domain tasks, as shown in Figure 2. Specifically, we investigate the impact of three key components: (i) the domain-aware guidance network, which dynamically steers the diffusion trajectory; (ii) MMD alignment, which matches marginal distributions between domains; and (iii) adjacency constraints, which preserve local structural information during transfer. The results reveal that the full model achieves the best performance across all tasks,

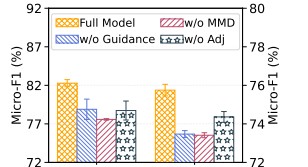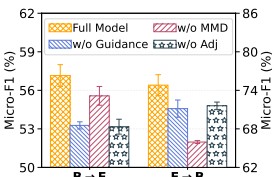

Figure 2: Classification Mi-F1 comparisons between DiffGDA variants on four cross-domain tasks.

confirming the effectiveness of integrating all three components. Removing the guidance network leads to substantial drops in performance, especially on the more challenging tasks (e.g., $\mathbf{B} \rightarrow \mathbf{E}$), highlighting its role in maintaining task-relevant diffusion dynamics. MMD alignment has a more pronounced effect on simpler shifts such as in $\mathbf{A} \rightarrow \mathbf{C}$ transfer task, where distribution-level matching is crucial. The removal of adjacency constraints consistently degrades performance, indicating their utility in preserving structural dependencies. Overall, each module plays a complementary role: the guidance network stabilizes the diffusion path, MMD facilitates cross-domain alignment, and adjacency constraints reinforce relational structure.

Additional ablation results are provided in Appendix E.3.

## 5.4 HYPER-PARAMETER ANALYSIS (**RQ3**)

In this section, we analyze the impact of three key parameters on the proposed DiffGDA framework, as illustrated in Figure 3: the sample ratio $\alpha$, the loss weight $\eta$, and the diffusion step $\mathsf{T}$.

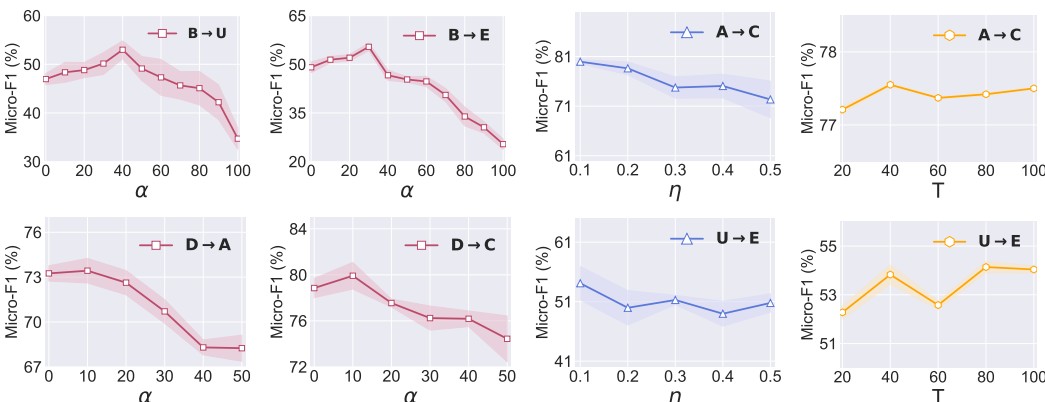

Figure 3: The performances of our DiffGDA w.r.t varying $\alpha$, $\eta$ and $\mathsf{T}$ on different transfer tasks.

**Study on Sample Ratio $\alpha$.** The results demonstrate the impact of varying subgraph sampling sizes on the Mi-F1 performance across different transfer tasks. For the first two tasks ($\mathbf{B} \rightarrow \mathbf{U}$ and $\mathbf{B} \rightarrow \mathbf{E}$), the model was evaluated across the full range of sampling sizes (0 ~100%), revealing a trend where performance initially improves with larger subgraphs, as critical structural information is preserved, but may plateau or decline beyond a certain point due to computational overhead or over-smoothing. In contrast, for the latter tasks, the sampling range was truncated to 0 ~ 50% due to out-of-memory issues, indicating that these tasks involve larger or more complex graphs where aggressive subgraph sampling is necessary to maintain feasibility. This aligns with the model's design goal of leveraging diffusion-based generation while managing scalability. Further investigation is needed to determine the relationship between graph properties and the ideal sampling size.

**Study on Loss Weight $\eta$.** We analyze the impact of coefficient $\eta$ in Eq. (14). As shown in Figure 3, both tasks exhibit a consistent decline in Mi-F1 as $\eta$ increases. This suggests that stronger alignment (i.e., larger $\eta$) may induce excessive regularization, hindering the model's ability to preserve task-specific discriminative features. Additional results are provided in Appendix E.4.

**Study on Diffusion Step $\mathsf{T}$.** As shown in Figure 3, the Mi-F1 score for $\mathbf{A} \rightarrow \mathbf{C}$ improves with increasing $\mathsf{T}$ and stabilizes beyond $\mathsf{T} = 40$, suggesting that a sufficient number of diffusion steps is necessary to effectively capture structural dependencies. A similar trend is observed in $\mathbf{U} \rightarrow \mathbf{E}$, where performance peaks at $\mathsf{T} = 80$ and fluctuates slightly thereafter. These results indicate that while adequate diffusion depth is important, excessively large $\mathsf{T}$ yields diminishing returns. Additional results across datasets are provided in Appendix E.4.

## 5.5 MODEL EFFICIENCY ANALYSIS (**RQ4**)

Table 3 compares the runtime performance of four approaches: UDAGCN, DANE, GraphAlign and DiffGDA on two tasks, $\mathbf{A} \rightarrow \mathbf{D}$ and $\mathbf{A} \rightarrow \mathbf{C}$, with all models trained for 150 epochs. Addition-

| Model | A → D | | | | A → C | | | |
|---|---|---|---|---|---|---|---|---|
| | Generating | Training | Total | Mi-F1 | Generating | Training | Total | Mi-F1 |
| UDAGCN | - | 83.19 | **83.19** | 67.52 | - | 104.91 | **104.91** | 72.64 |
| DANE | - | 134.92 | 134.92 | 62.41 | - | 167.00 | 167.00 | 69.77 |
| GraphAlign | 250.13 | 19.52 | 269.65 | 70.14 | 274.86 | 22.29 | 297.15 | 76.62 |
| DiffGDA | 103.98 | 22.75 | 126.73 | **73.41** | 102.31 | 23.10 | 125.41 | **80.23** |

Table 3: Runtime (seconds) and Mi-F1 (%) comparison on two transfer tasks. All models are trained using a single RTX 4090 GPU.

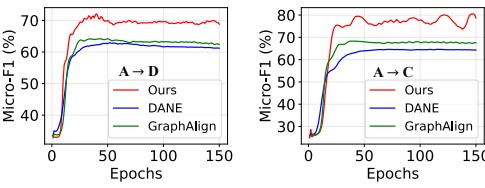

Figure 5: The training curves of Mi-F1 scores on three baselines across two cross-domain transfer scenarios over 150 epochs.

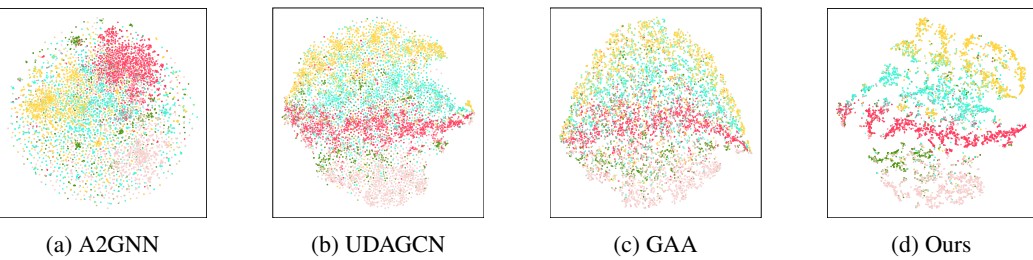

(a) A2GNN  (b) UDAGCN  (c) GAA  (d) Ours

Figure 6: Visualization of node embeddings inferred by four models for the **A → C** task.

ally, the training progress is visualized in Figure 5. While our method does not have the lowest absolute runtime, it achieves a relatively lower total runtime compared to GraphAlign, a state-of-the-art approach for GDA based on graph generation, which exhibits significant time overhead due to intermediate graph generation. As shown in Table 3, our method effectively strikes a balance between graph generation and training time, reducing the total runtime by over 50% compared to GraphAlign, with a smaller training time and a manageable graph generation cost. At the same time, it achieves superior Mi-F1 scores on both tasks. Furthermore, Figure 5 highlights that our approach not only converges faster but also consistently outperforms the baselines in terms of Mi-F1 scores across epochs. These results collectively demonstrate that our diffusion-based method offers a compelling trade-off between computational efficiency and performance, making it a suitable choice for cross-domain tasks that demand rigorous experimentation and reliable outcomes.

## 5.6 VISUALIZATION FOR REPRESENTATION SPACE (**RQ5**)

To qualitatively assess the learned node representations, we visualize their distributions using t-SNE (Maaten & Hinton, 2008; Yuan et al., 2025), as shown in Figure 6. In the plot, each color represents a distinct class, enabling an intuitive evaluation of the clustering quality. Our method, DiffGDA, produces more compact and well-separated clusters compared to other models, especially the strong data-driven baseline, GAA. This improved clustering suggests a more effective separation of the underlying class distributions within the target domain. These results demonstrate that our diffusion-based adaptation process effectively mitigates domain-irrelevant noise, enhancing inter-class discriminability and leading to a cleaner, more structured representation of the target graph.

## 6 CONCLUSION

In this work, we revisit the problem of Graph Domain Adaptation (GDA) from a continuous generative perspective. By framing cross-domain adaptation as a stochastic diffusion process, our proposed method, DiffGDA, effectively models the joint evolution of structure and semantics in continuous time. This continuous formulation naturally captures smooth structural transitions and avoids the rigid assumptions of discrete alignment frameworks. The incorporation of a guidance network enables the model to automatically learn the optimal adaptation path. Extensive empirical results across 14 transfer tasks validate the effectiveness and superiority of our approach.

## ACKNOWLEDGEMENTS

This work was supported by the National Key Research and Development Program of China under Grant Nos. 2024YFF0729003, the National Natural Science Foundation of China under Grant Nos. 62176014, 62276015, 62206266, the Fundamental Research Funds for the Central Universities.

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

# Appendix

## Table of Contents

# A    THEORETICAL PROOF

## A.1    PROOF OF THEOREM 1

Let $\mathbf{G}^{\mathcal{S}}$ and $\mathbf{G}^{\mathcal{T}}$ denote the source graph and target graph, respectively. Assume that the data distributions $p$ and $q$ define the forward diffusion processes on the source and target domains, respectively. For consistency of modeling, we consider the forward process in the target domain to share the same functional form as that in the source domain, i.e., $p(\mathbf{G}_0^{\mathcal{S}}|\mathbf{G}_t^{\mathcal{S}}) = q(\mathbf{G}_0^{\mathcal{T}}|\mathbf{G}_t^{\mathcal{T}})$. Under this formulation, the optimal diffusion network $\mathbb{P}(\ell^\star)$ for the target graph $\mathbf{G}^{\mathcal{T}}$ satisfies:

$$\mathbb{P}(\ell^\star) = \nabla_{\mathbf{G}_t^{\mathcal{S}}} \log p_t(\mathbf{G}_t^{\mathcal{S}}) + \nabla_{\mathbf{G}_t^{\mathcal{S}}} \log \mathbb{E}_{p(\mathbf{G}_0^{\mathcal{S}}|\mathbf{G}_t^{\mathcal{S}})} \frac{q(\mathbf{G}_0^{\mathcal{T}})}{p(\mathbf{G}_0^{\mathcal{S}})}, \tag{16}$$

where the first term is score function of the source graph, and the second term represents the gradient of the density ratio between the target and source graph distributions. The density ratio is defined as the likelihood ratio between the target and source domains in latent space. It reflects how domain-specific a given representation is, and provides a directional signal to guide diffusion process.

**Remark on Consistent Diffusion.** In this theorem, we assume that clean graph data are corrupted in the same way in the source and target domains. Although the data distributions in the two domains $(p(\mathbf{G}_0^{\mathcal{S}})$ and $q(\mathbf{G}_0^{\mathcal{T}}))$ are different, we assume they follow the same physical diffusion model (e.g., the same variance-exploding (VE) SDE). This is a common simplifying assumption that allows us to embed both domains into the same diffusion latent space for operations and comparisons. Based on these shared diffusion dynamics, inter-domain differences are entirely reflected in the differences of their initial-state ($t = 0$) data distributions, and the core task of our guidance network $\mathbb{Q}$ is to estimate and compensate for this discrepancy via the density ratio $q(\mathbf{G}_0^{\mathcal{T}})/p(\mathbf{G}_0^{\mathcal{S}})$.

**Proof:** For clarity in the proof, we denote the left-hand side and right-hand side of Eq. (16) as **LE** and **RE**, respectively. Our goal is to prove the equivalence by transforming **RE** into **LE**. In the first step, we establish a connection between score matching on the target graph and importance-weighted denoising score matching on the source graph. Drawing inspiration from (Vincent, 2011; Ouyang et al., 2024), this relationship can be formalized as:

$$\ell^\star = \arg\min_{\ell} \mathbb{E}_t \left\{ \lambda_t \mathbb{E}_{q_t(\mathbf{G}_t^{\mathcal{T}})} \left[ \left\| \mathbb{P}(\ell) - \nabla_{\mathbf{G}_t^{\mathcal{T}}} \log q_t(\mathbf{G}_t^{\mathcal{T}}) \right\|_2^2 \right] \right\}$$

$$\implies \arg\min_{\ell} \mathbb{E}_t \left\{ \lambda_t \mathbb{E}_{p(\mathbf{G}_0^{\mathcal{S}})} \mathbb{E}_{p(\mathbf{G}_t^{\mathcal{S}}|\mathbf{G}_0^{\mathcal{S}})} \left[ \left\| \mathbb{P}(\ell) - \nabla_{\mathbf{G}_0^{\mathcal{T}}} \log p(\mathbf{G}_t^{\mathcal{S}}|\mathbf{G}_0^{\mathcal{S}}) \right\|_2^2 \frac{q(\mathbf{G}_0^{\mathcal{T}})}{p(\mathbf{G}_0^{\mathcal{S}})} \right] \right\}. \tag{17}$$

The first term corresponds to score matching on source graph, while the second term incorporates importance weighting, bridging the connection to denoising score matching on the source graph. This adjustment is modulated by $\frac{q(\mathbf{G}_0^{\mathcal{T}})}{p(\mathbf{G}_0^{\mathcal{S}})}$, which accounts for the distributional differences between the source and target graphs, thereby ensuring a consistent and unbiased score estimation.

Leveraging importance-weighted denoising score matching on the source graph, the optimal score can be formulated as:

$$\mathbb{P}(\ell^\star) = \frac{\mathbb{E}_{p(\mathbf{G}_0^{\mathcal{S}}|\mathbf{G}_t^{\mathcal{S}})} \left[ \frac{q(\mathbf{G}_0^{\mathcal{T}})}{p(\mathbf{G}_0^{\mathcal{S}})} \nabla_{\mathbf{G}_t^{\mathcal{S}}} \log p(\mathbf{G}_t^{\mathcal{S}}|\mathbf{G}_0^{\mathcal{S}}) \right]}{\mathbb{E}_{p(\mathbf{G}_0^{\mathcal{S}}|\mathbf{G}_t^{\mathcal{S}})} \left[ \frac{q(\mathbf{G}_0^{\mathcal{T}})}{p(\mathbf{G}_0^{\mathcal{S}})} \right]} = \mathbf{LE}. \tag{18}$$

The final optimal solution $\mathbb{P}(\ell^\star)$ is computed by normalizing the weighted gradient of the source graph's log-probability. The numerator captures the weighted score evaluation, while the denominator ensures proper scaling by accounting for the overall importance weights. This formulation guarantees an unbiased and consistent estimation, as the importance weights $q(G_0^T)/p(G_0^S)$ correct for any domain-level discrepancies.

In the second step, without loss of generality, we consider an arbitrary graph $\mathbf{G}$ within the framework of the diffusion process $p$ and proceed with the following derivation:

$$\nabla_{\mathbf{G}_t} \log p(\mathbf{G}_0|\mathbf{G}_t) = \nabla_{\mathbf{G}_t} \log p(\mathbf{G}_t|\mathbf{G}_0) - \nabla_{\mathbf{G}_t} \log p_t(\mathbf{G}_t). \tag{19}$$

This expression establishes the relationship between the conditional score $\nabla_{\mathbf{G}_t} \log p(\mathbf{G}_0|\mathbf{G}_t)$ and the components of the forward diffusion process, specifically the reverse transition probability and the marginal distribution at time $t$.

Finally, based on Eq. (19), we can rewrite Eq. (16) as:

$$
\begin{aligned}
\mathbf{RE} &= \nabla_{\mathbf{G}_t^{\mathcal{S}}} \log p_t(\mathbf{G}_t^{\mathcal{S}}) + \nabla_{\mathbf{G}_t^{\mathcal{S}}} \log \mathbb{E}_{p(\mathbf{G}_t^{\mathcal{S}}|\mathbf{G}_0^{\mathcal{S}})} \left[ \frac{q(\mathbf{G}_0^{\mathcal{T}})}{p(\mathbf{G}_0^{\mathcal{S}})} \right] \\
&= \nabla_{\mathbf{G}_t^{\mathcal{S}}} \log p_t(\mathbf{G}_t^{\mathcal{S}}) + \frac{\mathbb{E}_{p(\mathbf{G}_0^{\mathcal{S}}|\mathbf{G}_t^{\mathcal{S}})} \left[ \frac{q(\mathbf{G}_0^{\mathcal{T}})}{p(\mathbf{G}_0^{\mathcal{S}})} \nabla_{\mathbf{G}_t^{\mathcal{S}}} \log p(\mathbf{G}_t^{\mathcal{S}}|\mathbf{G}_0^{\mathcal{S}}) \right]}{\mathbb{E}_{p(\mathbf{G}_0^{\mathcal{S}}|\mathbf{G}_t^{\mathcal{S}})} \left[ \frac{q(\mathbf{G}_0^{\mathcal{T}})}{p(\mathbf{G}_0^{\mathcal{S}})} \right]} \\
&= \nabla_{\mathbf{G}_t^{\mathcal{S}}} \log p_t(\mathbf{G}_t^{\mathcal{S}}) + \frac{\mathbb{E}_{p(\mathbf{G}_0^{\mathcal{S}}|\mathbf{G}_t^{\mathcal{S}})} \left[ \frac{q(\mathbf{G}_0^{\mathcal{T}})}{p(\mathbf{G}_0^{\mathcal{S}})} \nabla_{\mathbf{G}_t^{\mathcal{S}}} \log p(\mathbf{G}_t^{\mathcal{S}}|\mathbf{G}_0^{\mathcal{S}}) \right]}{\mathbb{E}_{p(\mathbf{G}_0^{\mathcal{S}}|\mathbf{G}_t^{\mathcal{S}})} \left[ \frac{q(\mathbf{G}_0^{\mathcal{S}})}{p(\mathbf{G}_0^{\mathcal{S}})} \right]} - \nabla_{\mathbf{G}_t^{\mathcal{S}}} \log p_t(\mathbf{G}_t^{\mathcal{S}}) \\
&= \frac{\mathbb{E}_{p(\mathbf{G}_0^{\mathcal{S}}|\mathbf{G}_t^{\mathcal{S}})} \left[ \frac{q(\mathbf{G}_0^{\mathcal{T}})}{p(\mathbf{G}_0^{\mathcal{S}})} \nabla_{\mathbf{G}_t^{\mathcal{S}}} \log p(\mathbf{G}_t^{\mathcal{S}}|\mathbf{G}_0^{\mathcal{S}}) \right]}{\mathbb{E}_{p(\mathbf{G}_0^{\mathcal{S}}|\mathbf{G}_t^{\mathcal{S}})} \left[ \frac{q(\mathbf{G}_0^{\mathcal{T}})}{p(\mathbf{G}_0^{\mathcal{S}})} \right]} = \mathbf{LE}.
\end{aligned}
\tag{20}
$$

Therefore, we have proven that Theorem 1 holds.

### A.2  PROOF OF GUIDANCE NETWORK $\mathbb{Q}$

In this section, we demonstrate that Eq. (13) holds, namely:

$$
\mathbb{Q}(\boldsymbol{\delta}^{\star}) = (\mathbb{Q}(\boldsymbol{\delta}_1^{\star}), \mathbb{Q}(\boldsymbol{\delta}_2^{\star})) = \log \mathbb{E}_{p(\mathbf{G}_0^{\mathcal{S}}|\mathbf{G}_t^{\mathcal{S}})} \frac{q(\mathbf{G}_0^{\mathcal{T}})}{p(\mathbf{G}_0^{\mathcal{S}})},
\tag{21}
$$

where $\mathbb{Q}(\boldsymbol{\delta})$ denotes a learned representation parameterized by $\boldsymbol{\delta}$, and the logarithm is applied element-wise. To proceed, let us define $p(\mathbf{G}_0^{\mathcal{S}}) = p(\mathbf{X}_0^{\mathcal{S}}, \mathbf{A}_0^{\mathcal{S}})$ and $q(\mathbf{G}_0^{\mathcal{T}}) = q(\mathbf{X}_0^{\mathcal{T}}, \mathbf{A}_0^{\mathcal{T}})$ as the source and target joint distributions over node features $\mathbf{X}$ and adjacency matrices $\mathbf{A}$, respectively. Note that, for simplicity of representation, we use $\mathbf{G}$ as a shorthand for $(\mathbf{X}, \mathbf{A})$ throughout this proof. Thus, we have:

$$
\begin{aligned}
\boldsymbol{\delta}^{\star} &= \arg\min_{\boldsymbol{\delta}} \int_{\mathbf{G}_t^{\mathcal{S}}} \left\| \mathbb{Q}(\boldsymbol{\delta}) - \mathbb{E}_{p(\mathbf{G}_0^{\mathcal{S}}|\mathbf{G}_t^{\mathcal{S}})} \left[ \frac{q(\mathbf{G}_0^{\mathcal{T}})}{p(\mathbf{G}_0^{\mathcal{S}})} \right] \right\|_2^2 p(\mathbf{G}_t^{\mathcal{S}}) \, d\mathbf{G}_t^{\mathcal{S}} \\
&= \arg\min_{\boldsymbol{\delta}} \int_{\mathbf{G}_t^{\mathcal{S}}} \left\| \mathbb{Q}(\boldsymbol{\delta}) - \int_{\mathbf{G}_0^{\mathcal{S}}} p(\mathbf{G}_0^{\mathcal{S}}|\mathbf{G}_t^{\mathcal{S}}) \cdot \frac{q(\mathbf{G}_0^{\mathcal{T}})}{p(\mathbf{G}_0^{\mathcal{S}})} \, d\mathbf{G}_0^{\mathcal{S}} \right\|_2^2 p(\mathbf{G}_t^{\mathcal{S}}) \, d\mathbf{G}_t^{\mathcal{S}} \\
&= \arg\min_{\boldsymbol{\delta}} \int_{\mathbf{G}_t^{\mathcal{S}}} \left\{ \|\mathbb{Q}(\boldsymbol{\delta})\|_2^2 - 2 \left\langle \mathbb{Q}(\boldsymbol{\delta}), \int_{\mathbf{G}_0^{\mathcal{S}}} p(\mathbf{G}_0^{\mathcal{S}}|\mathbf{G}_t^{\mathcal{S}}) \cdot \frac{q(\mathbf{G}_0^{\mathcal{T}})}{p(\mathbf{G}_0^{\mathcal{S}})} \, d\mathbf{G}_0^{\mathcal{S}} \right\rangle \right\} p(\mathbf{G}_t^{\mathcal{S}}) \, d\mathbf{G}_t^{\mathcal{S}} + C \\
&= \arg\min_{\boldsymbol{\delta}} \iint \left\| \mathbb{Q}(\boldsymbol{\delta}) - \frac{q(\mathbf{G}_0^{\mathcal{T}})}{p(\mathbf{G}_0^{\mathcal{S}})} \right\|_2^2 \cdot p(\mathbf{G}_0^{\mathcal{S}}|\mathbf{G}_t^{\mathcal{S}}) \cdot p(\mathbf{G}_t^{\mathcal{S}}) \, d\mathbf{G}_0^{\mathcal{S}} d\mathbf{G}_t^{\mathcal{S}} \\
&= \arg\min_{\boldsymbol{\delta}} \iint \left\| \mathbb{Q}(\boldsymbol{\delta}) - \frac{q(\mathbf{G}_0^{\mathcal{T}})}{p(\mathbf{G}_0^{\mathcal{S}})} \right\|_2^2 \cdot p(\mathbf{G}_0^{\mathcal{S}}, \mathbf{G}_t^{\mathcal{S}}) \, d\mathbf{G}_0^{\mathcal{S}} d\mathbf{G}_t^{\mathcal{S}} \\
&= \arg\min_{\boldsymbol{\delta}} \mathbb{E}_{p(\mathbf{G}_0^{\mathcal{S}}, \mathbf{G}_t^{\mathcal{S}})} \left\| \mathbb{Q}(\boldsymbol{\delta}) - \frac{q(\mathbf{G}_0^{\mathcal{T}})}{p(\mathbf{G}_0^{\mathcal{S}})} \right\|_2^2,
\end{aligned}
\tag{22}
$$

where $C$ is a constant independent of $\boldsymbol{\delta}$.

According to Eq. (12):

$$
\boldsymbol{\delta}^{\star} = (\boldsymbol{\delta}_1^{\star}, \boldsymbol{\delta}_2^{\star}) = \arg\min_{\boldsymbol{\delta}} \mathbb{E}_{p(\mathbf{G}_0^{\mathcal{S}}, \mathbf{G}_t^{\mathcal{S}})} \left\| \mathbb{Q}(\boldsymbol{\delta}) - \frac{q(\mathbf{G}_0^{\mathcal{T}})}{p(\mathbf{G}_0^{\mathcal{S}})} \right\|_2^2.
\tag{23}
$$

Thereby we complete the proof.

# B  IMPLEMENTATION DETAILS

---

**Algorithm 1** Density Ratio Estimation

---

**Require:** Source graph distribution $p(\mathbf{G}_0^{\mathcal{S}})$, target graph distribution $q(\mathbf{G}_0^{\mathcal{T}})$, pre-trained domain classifier $\mathcal{C}_{gnn}$, number of Monte Carlo samples $S_{mc}$
**Ensure:** Estimated density ratio: $\frac{r_1}{S_{mc}}$
1: Train the domain classifier $\mathcal{C}_{gnn}$ to distinguish $\mathbf{G}_0^{\mathcal{S}}$ and $\mathbf{G}_0^{\mathcal{T}}$
2: Initialize accumulator: $r_1 \leftarrow 0$
3: **for** $i = 1$ to $S_{mc}$ **do**
4:     Sample $\mathbf{G}_0^{\mathcal{S}}(i) \sim p(\mathbf{G}_0^{\mathcal{S}})$
5:     Sample $\mathbf{G}_0^{\mathcal{T}(i)} \sim q(\mathbf{G}_0^{\mathcal{T}})$
6:     Compute $\mathbf{y}^{(i)} = \mathcal{C}_{gnn}(\mathbf{G}_0^{\mathcal{S}(i)}, \mathbf{G}_0^{\mathcal{T}(i)})$         ▷ For each node in the graphs
7:     Calculate ratio: $r^{(i)} = \frac{1-\mathbf{y}^{(i)}}{\mathbf{y}^{(i)}}$
8:     Accumulate: $r_1 \leftarrow r_1 + r^{(i)}$
9: **end for**
10: **return** $\frac{r_1}{S_{mc}}$

---

# C  THEORETICAL ANALYSIS OF DIFFGDA'S ERROR BOUND

In this section, we analyze the theoretical error bound of DiffGDA framework, focusing on the factors that influence its performance under GDAsettings. By extending the traditional error bound framework and incorporating components specific to diffusion-based methods, we derive an expression that characterizes the generalization error on the target domain as a function of multiple factors, including source domain performance, domain discrepancy, sample complexity, and noise stability.

## C.1  ERROR BOUND EXPRESSION

Inspired by (Huang et al., 2024), the generalization error on target domain $\mathcal{E}_T$ can be bounded as:

$$\mathcal{E}_{\mathcal{T}} \le \mathcal{E}_{\mathcal{S}} + \xi \cdot \mathcal{D}(\mathcal{P}_{\mathcal{S}}, \mathcal{P}_{\mathcal{T}}) + \frac{C}{\sqrt{N}} + \mu \cdot \mathcal{R}_{\text{diff}}, \tag{24}$$

where $\mathcal{E}_{\mathcal{T}}$ is the generalization error on the target domain, $\mathcal{E}_S$ is the empirical error on the source domain, and $\mathcal{D}(\mathcal{P}_{\mathcal{S}}, \mathcal{P}_{\mathcal{T}})$ represents the distributional discrepancy between the source and target domains, which can be measured using metrics such as MMD or Wasserstein distance. The term $\xi$ is a weight factor that scales the impact of domain discrepancy, influenced by the model's capacity and alignment mechanism. The term $\frac{C}{\sqrt{N}}$ reflects a sample complexity term, where $N$ represents the number of source domain samples, and $C$ denotes model complexity (e.g., Lipschitz constant or parameter count). Finally, $\mathcal{R}_{\text{diff}}$ captures the residual error introduced by the diffusion process, which can be further decomposed into three controllable sources:

$$\mathcal{R}_{\text{diff}} = \mu_1 \, \mathcal{B}_{\text{sched}}(\sigma_{\min}, \sigma_{\max}) + \mu_2 \, \mathcal{B}_{\text{disc}}(\mathsf{T}) + \mu_3 \, \varepsilon_{\text{ratio}}, \tag{25}$$

where $\eta_1, \eta_2, \eta_3$ are non-negative coefficients weighting each term; $\mathcal{B}_{\text{sched}}(\sigma_{\min}, \sigma_{\max})$ denotes the error from the variance-exploding (VE) noise schedule, controlled by the minimum and maximum noise levels $\sigma_{\min}$ and $\sigma_{\max}$. $\mathcal{B}_{\text{disc}}(\mathsf{T})$ is the discretization error of the reverse diffusion process, which decreases as the number of reverse steps $\mathsf{T}$ increases. And $\varepsilon_{\text{ratio}}$ is the approximation error of the density-ratio guidance network, capturing the gap between estimated and true density ratios.

## C.2  ANALYSIS AND INSIGHTS

The above error bound provides a comprehensive understanding of the factors that contribute to the generalization ability of DiffGDA in the target domain:

**1. Source Domain Error $\mathcal{E}_{\mathcal{S}}$:** This error serves as the foundation of the target domain performance. A lower $\mathcal{E}_{\mathcal{S}}$ indicates that the diffusion model effectively captures the structural and semantic information in the source domain, which can be transferred to the target domain. If $\mathcal{E}_{\mathcal{S}}$ is large, it becomes challenging to bridge the domain gap, even with a well-designed alignment strategy.

**2. Domain Discrepancy** $\mathcal{D}(\mathcal{P}_{\mathcal{S}}, \mathcal{P}_{\mathcal{T}})$**:** This term reflects the divergence between the source and target distributions. By leveraging the domain-aware guidance network in DiffGDA, the model minimizes $\mathcal{D}(\mathcal{P}_{\mathcal{S}}, \mathcal{P}_{\mathcal{T}})$ through adaptive alignment in the latent space. The density ratio term $\log \frac{q(\mathbf{G}_0^{\mathcal{T}})}{p(\mathbf{G}_0^{\mathcal{S}})}$ plays a pivotal role in reducing this discrepancy, ensuring that the generated graphs align closely with the target distribution.

**3. Sample Complexity** $\frac{C}{\sqrt{N}}$**:** This term highlights the importance of having sufficient labeled samples in the source domain. As $N$ increases, the error bound tightens, indicating better generalization. However, larger models (with higher $C$) may require more samples to avoid overfitting. DiffGDA mitigates this by efficiently generating intermediate graphs that maximize the utility of source data.

**4. Diffusion Residual** $\mathcal{R}_{\text{diff}}$**:** The residual of the diffusion process is governed by three sources: (i) the noise schedule $\mathcal{B}_{\text{sched}}(\sigma_{\min}, \sigma_{\max})$, which controls the stability of forward corruption and reverse denoising; (ii) the discretization error $\mathcal{B}_{\text{disc}}(\mathsf{T})$, which decreases as more reverse steps $\mathsf{T}$ are used; and (iii) the density-ratio approximation error $\varepsilon_{\text{ratio}}$, which depends on the accuracy of the guidance network. Each of these terms provides a direct knob for improving error control.

## C.3 Practical Implications

The error bound highlights three key aspects that influence the success of DiffGDA:

- **Domain Alignment:** Reducing $\mathcal{D}(\mathcal{P}_{\mathcal{S}}, \mathcal{P}_{\mathcal{T}})$ through adaptive guidance and density ratio estimation is critical for effective knowledge transfer.
- **Sample Efficiency:** The ability to leverage limited source domain samples is vital, especially in low-resource scenarios.
- **Diffusion Stability:** Properly tuning the noise schedule, the number of reverse steps, and the accuracy of density-ratio guidance jointly ensures robustness and lowers the residual error in GDA.

In summary, this error bound clarifies that the diffusion residual is not a monolithic term but decomposes into three controllable sources, directly linked to design choices in DiffGDA. Optimizing these factors further improves the theoretical and practical guarantees of cross-domain performance.

## D EXPERIMENT SETUP

### D.1 DATASET DESCRIPTION

We perform node classification experiments under three migration settings in Table 2. In each scenario, we train our model on one graph and evaluate it on other graphs.

- ACMv9 (**A**), Citationv1 (**C**), DBLPv7 (**D**): These datasets are citation networks from different sources, where each node represents a research paper, and each edge represents a citation relationship between two papers. The task is to predict the category of each research paper. The data are collected from ACM (before 2008), DBLP (2004-2008), and Microsoft Academic Graph (after 2010). We set six transfer scenarios: $\mathbf{A} \to \mathbf{C}$, $\mathbf{A} \to \mathbf{D}$, $\mathbf{C} \to \mathbf{A}$, $\mathbf{C} \to \mathbf{D}$, $\mathbf{D} \to \mathbf{A}$, and $\mathbf{D} \to \mathbf{C}$.
- USA (**U**), Brazil (**B**), Europe (**E**): These datasets are collected from transportation statistics and primarily contain aviation activity data, where each node represents an airport, and each edge represents a flight connection between two airports. The task is to predict the category of each airport. We set six transfer scenarios: $\mathbf{U} \to \mathbf{B}$, $\mathbf{U} \to \mathbf{E}$, $\mathbf{B} \to \mathbf{U}$, $\mathbf{B} \to \mathbf{E}$, $\mathbf{E} \to \mathbf{U}$, and $\mathbf{E} \to \mathbf{B}$.
- Blog1 (**B1**), Blog2 (**B2**): These datasets are taken from the BlogCatalog dataset, where nodes represent bloggers, and edges represent the friendship relationships between bloggers. The node attributes consist of keywords extracted from the bloggers' self-descriptions. The task is to predict the corresponding group to which they belong. We set two scenarios: $\mathbf{B1} \to \mathbf{B2}$ and $\mathbf{B2} \to \mathbf{B1}$.

### D.2 BASELINES

We compare our method with the following baseline methods, which can be divided into two categories: (1) Graph Neural Networks (GNNs) on the source graph only: GAT (Veličković et al., 2018), GIN (Xu et al., 2019), and GCN (Kipf & Welling, 2017) are trained end-to-end on the source graph,

allowing direct application to the target graph; (2) Graph Domain Adaptation Methods: DANE (Zhang et al., 2019), UDAGCN (Wu et al., 2020), AdaGCN (Dai et al., 2022), StruRW (Liu et al., 2023a), GRADE (Wu et al., 2023), PairAlign (Liu et al., 2024b), GraphAlign (Huang et al., 2024), A2GNN (Liu et al., 2024a), GGDA (Lei et al., 2025), TDSS (Chen et al., 2025a), DGSDA (Yang et al., 2025) and GAA (Fang et al., 2025a) are specifically designed to address the GDA problem.

Below, we provide descriptions of the baselines used in the experiments.

- **GAT , GIN , GCN** , train using GAT, GIN and GCN under standard ERM.
- **DANE** uses shared weight GCNs to get node representations and then handles distribution shift via least square generative adversarial network.
- **UDAGCN** is a model-centric method that first combines adversarial learning with a GNN model.
- **AdaGCN** is a model-centric method for GDA. It utilizes the techniques of adversarial domain adaptation with graph convolution, and employs the empirical Wasserstein distance between the source and target distributions of node representation as the regularization.
- **STRURW** modifies graph data by adjusting edge weights, while the modification relies on the training of data-centric GDA methods.
- **GRADE** introduces graph subtree discrepancy as a metric to measure the graph distribution shift by connecting GNNs with WL subtree kernel.
- **PairAlign** not only recalibrates the influence between neighboring nodes using edge weights to address conditional structure shifts but also adjusts the classification loss with label weights to account for label shifts.
- **GraphAlign** generates a small, transferable graph from the source graph, which is used for GNN training with Empirical Risk Minimization (ERM).
- **A2GNN** proposes a simple GNN which stacks more propagation layers on target graph.
- **GGDA** using the Fused Gromov-Wasserstein (FGW) metric and adaptive vertex selection, ensures effective knowledge transfer between source and target domains.
- **TDSS** applies structural smoothing directly to the target graph to reduce local variations and enhance model robustness against distribution shifts. TDSS improves knowledge transfer by preserving node representation consistency while mitigating the impact of structural discrepancies.
- **DGSDA** addresses GDA by disentangling node attribute and graph topology alignment. It employs Bernstein polynomial approximation for efficient spectral filter alignment, avoiding costly eigenvalue decomposition, and enables flexible GNNs for improved cross-domain adaptability.
- **GAA** tackles both graph topology and node attribute shifts using an attention-based cross-channel module for effective alignment, enhancing the model's focus on critical features.

### D.3  IMPLEMENTATION DETAILS

We evaluate the model performance using Mi-F1 and Ma-F1 scores. The hidden layer dimension is set to 64, and the learning rate is fine-tuned within the set $\{0.0001, 0.001, 0.01\}$ to achieve optimal performance. Domain alignment is implemented using the MMD method with a Gaussian kernel (Keerthi & Lin, 2003). To prevent overfitting, a dropout rate of 0.2 is applied. We adjust the parameters $\alpha$ within the range $[0, 1]$, $\eta$ within the range $[0, 0.5]$, and $\mathsf{T}$ within the range $[0, 150]$. All experiments are conducted using PyTorch on an NVIDIA RTX 4090 GPU, running for 5 rounds with 150 epochs per round. For each round, the maximum Mi-F1 and Ma-F1 scores are recorded, and their mean and variance are calculated.

In our diffusion models, we adopt the Variance Exploding (VE) SDE, defined by the following stochastic differential equation:

$$dx = \sigma_{\min}\left(\frac{\sigma_{\max}}{\sigma_{\min}}\right)^t \sqrt{2\log\left(\frac{\sigma_{\max}}{\sigma_{\min}}\right)}dw, \tag{26}$$

where $\sigma_{\min} = 0.001$, $\sigma_{\max} = 0.01$, and $t \in (0, 1]$. The transition probability density function $p_{t,t-\delta t}(x_{t-\delta t}|x_t)$ is a Gaussian distribution with mean $x_t$ and covariance $\Sigma_t$:

$$p_{t,t-\delta t}(x_{t-\delta t}|x_t) = \mathcal{N}(x_{t-\delta t}|x_t, \Sigma_t), \tag{27}$$

| Methods | Blog1 (**B1**), Blog2 (**B2**) | | | | |
| | **B1 → B2** | | **B2 → B1** | | **Avg.** |
| | Mi-F1 | Ma-F1 | Mi-F1 | Ma-F1 | |
|---|---|---|---|---|---|
| GAT (ICLR'18) | $21.15_{\pm2.72}$ | $11.18_{\pm3.84}$ | $19.46_{\pm1.41}$ | $9.43_{\pm3.04}$ | $15.30_{\pm2.75}$ |
| GIN (ICLR'19) | $20.51_{\pm2.16}$ | $11.44_{\pm2.68}$ | $18.60_{\pm0.38}$ | $9.57_{\pm0.28}$ | $15.03_{\pm1.38}$ |
| GCN (ICLR'17) | $23.17_{\pm1.10}$ | $13.80_{\pm1.15}$ | $23.46_{\pm0.48}$ | $13.76_{\pm1.33}$ | $18.55_{\pm1.02}$ |
| DANE (IJCAI'19) | $28.15_{\pm1.77}$ | $25.20_{\pm2.13}$ | $30.32_{\pm0.69}$ | $28.37_{\pm0.89}$ | $28.01_{\pm1.37}$ |
| UDAGCN (WWW'22) | $33.87_{\pm2.09}$ | $28.83_{\pm1.15}$ | $31.86_{\pm1.08}$ | $28.95_{\pm0.36}$ | $30.88_{\pm1.17}$ |
| AdaGCN (TKDE'22) | $30.75_{\pm0.64}$ | $20.69_{\pm1.40}$ | $27.17_{\pm2.34}$ | $18.73_{\pm2.91}$ | $24.34_{\pm1.82}$ |
| StruRW (ICML'23) | $40.37_{\pm0.44}$ | $39.62_{\pm0.77}$ | $42.01_{\pm0.44}$ | $41.36_{\pm0.47}$ | $40.84_{\pm0.53}$ |
| GRADE (AAAI'23) | $48.44_{\pm3.12}$ | $44.46_{\pm4.01}$ | $\underline{46.78}_{\pm1.31}$ | $42.00_{\pm1.52}$ | $45.42_{\pm2.49}$ |
| PairAlign (ICML'24) | $40.01_{\pm0.78}$ | $39.57_{\pm0.85}$ | $42.64_{\pm0.69}$ | $41.98_{\pm0.85}$ | $41.05_{\pm0.79}$ |
| GraphAlign (KDD'24) | $45.58_{\pm1.15}$ | $43.14_{\pm0.55}$ | $43.21_{\pm0.62}$ | $42.29_{\pm3.28}$ | $43.55_{\pm1.40}$ |
| A2GNN (AAAI'24) | $47.10_{\pm3.40}$ | $45.42_{\pm4.75}$ | $44.01_{\pm2.01}$ | $43.14_{\pm1.77}$ | $44.92_{\pm2.98}$ |
| GGDA (Arxiv'25) | $48.25_{\pm2.45}$ | $45.09_{\pm3.60}$ | $44.85_{\pm3.13}$ | $43.45_{\pm4.74}$ | $45.41_{\pm3.48}$ |
| TDSS (AAAI'25) | $\underline{49.53}_{\pm1.08}$ | $\underline{48.31}_{\pm2.00}$ | $44.20_{\pm0.29}$ | $\underline{44.31}_{\pm1.89}$ | $\underline{46.59}_{\pm1.31}$ |
| DGSDA (ICML'25) | $48.61_{\pm1.73}$ | $48.01_{\pm1.71}$ | $44.12_{\pm1.26}$ | $42.06_{\pm3.66}$ | $45.70_{\pm1.68}$ |
| GAA (ICLR'25) | $48.80_{\pm2.45}$ | $45.66_{\pm2.12}$ | $43.74_{\pm1.25}$ | $40.59_{\pm2.04}$ | $44.70_{\pm1.97}$ |
| **DiffGDA (Ours)** | $\mathbf{53.52}_{\pm2.93}$ | $\mathbf{51.18}_{\pm0.97}$ | $\mathbf{48.95}_{\pm2.48}$ | $\mathbf{47.17}_{\pm1.26}$ | $\mathbf{50.20}_{\pm1.91}$ |

Table 4: Node classification performance ($\% \pm \sigma$) on the Social domains. The highest scores are highlighted in **bold**, while the second-highest scores are underlined.

| Dataset | **A→C** | **C→A** | **A→D** |
|---|---|---|---|
| Baseline $-\log_2(0.05)$ | 4.32 | 4.32 | 4.32 |
| DiffGDA vs TDSS (AAAI'25) | 16.29 | 13.57 | 7.43 |
| DiffGDA vs AdaGCN (TKDE'22) | 30.84 | 26.46 | 19.57 |
| DiffGDA vs A2GNN (AAAI'24) | 31.98 | 18.96 | 23.67 |
| DiffGDA vs GraphAlign (KDD'24) | 45.39 | 43.36 | 40.63 |

Table 5: Significance test (t-test) of DiffGDA

where $\Sigma_t$ is given by:

$$\Sigma_t = \sigma_{\min}^2 \left(\frac{\sigma_{\max}}{\sigma_{\min}}\right)^{2t} - \sigma_{\min}^2 \left(\frac{\sigma_{\max}}{\sigma_{\min}}\right)^{2t-2\delta t}. \tag{28}$$

After generating samples by simulating the reverse diffusion process, we quantize the elements of the adjacency matrix to $\{0, 1\}$ by clipping values in the ranges $(-\infty, 3)$ to 0 and $[3, +\infty)$ to 1.

The guidance network is a 3-layer MLP with 512 hidden units and ReLU activation functions. We train the guidance network for 100 epochs.

# E  EXPERIMENT RESULTS

## E.1  MAIN EXPERIMENT RESULTS

The experimental results on Blog domains are presented in Table 4.

## E.2  SIGNIFICANCE TEST (T-TEST) OF DIFFGDA

To ensure that our method achieves statistically significant improvements, we conducted paired t-tests to compare the performance of DiffGDA against various baseline models across multiple transfer tasks, as presented in Table 5. This rigorous statistical analysis was carried out at the 0.05 significance level, allowing us to objectively assess the superiority of our approach. We leveraged $-\log_2(p)$ values to quantify statistical evidence, where higher $-\log_2(p)$ values indicate stronger support for the observed differences. By employing this approach, we aim to demonstrate its consistent and reliable performance across diverse tasks.

## E.3  ABLATION STUDY

The more ablation results are presented in Figure 7.

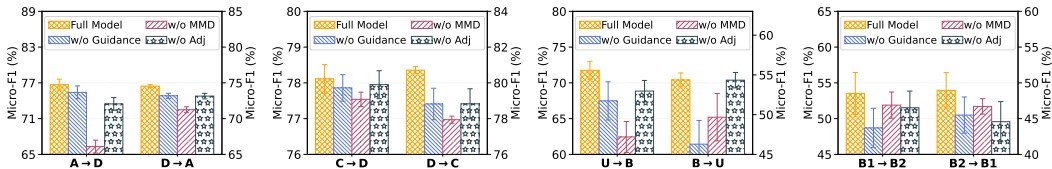

Figure 7: Classification Mi-F1 comparisons between DiffGDA variants on other cross-domain tasks.

## E.4 HYPER-PARAMETER ANALYSIS

The more hyper-parameter results are presented in Figure 8 and Figure 9.

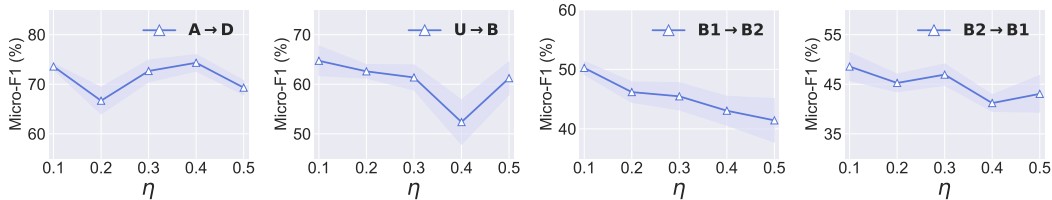

Figure 8: The performances of our DiffGDA w.r.t varying weight $\eta$ on different transfer tasks

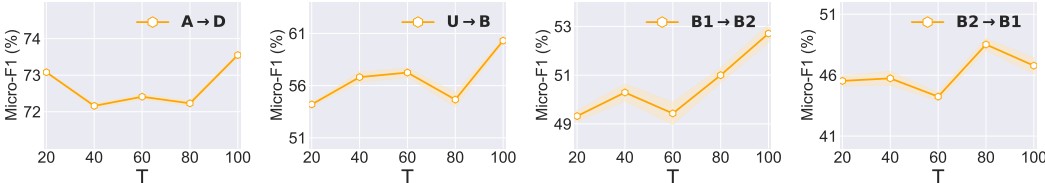

Figure 9: The performances of our DiffGDA w.r.t varying step $T$ on different transfer tasks

