# OpenReview forum: "Learning Structure-Semantic Evolution Trajectories for Graph Domain Adaptation"
_ICLR.cc/2026/Conference — ICLR 2026 Poster_

### Official Review · Reviewer_eQUv · 2025-10-31

**Soundness:** 3
**Presentation:** 4
**Contribution:** 3
**Rating:** 8
**Confidence:** 3

**Summary:**

This paper introduces DiffGDA, a novel approach for Graph Domain Adaptation (GDA) that frames the problem as a continuous-time generative process. Instead of relying on discrete alignment steps or the construction of intermediate graphs, DiffGDA models the evolution from the source to the target domain using stochastic differential equations (SDEs), i.e., a diffusion model. The core contributions include: (1) A formulation of GDA as a continuous-time process that jointly models structural and semantic (feature) transitions. (2) A "domain-aware guidance network" designed to steer the reverse diffusion process, effectively learning an adaptation path from the source to the target distribution. (3) A theoretical justification (Theorem 1) showing that this guidance mechanism, by learning the density ratio between domains, converges to an optimal adaptation trajectory. (4) Extensive experiments on 14 transfer tasks, demonstrating that DiffGDA consistently outperforms state-of-the-art GDA baselines.

**Strengths:**

1. The core idea of modeling GDA as a continuous, time-driven generative process is novel and compelling. It directly addresses a clear limitation of existing data-oriented methods, which often assume a discrete or linear transformation between domains. The argument that real-world graph evolution is continuous and nonlinear provides strong motivation for this diffusion-based approach.

2. The method is thoroughly evaluated against a wide array of recent GDA baselines (both model-oriented and data-oriented) across 8 datasets and 14 transfer tasks. The results in Tables 1 and 4 show that DiffGDA achieves state-of-the-art performance, often by a significant margin.

3. The paper provides a solid theoretical foundation for its proposed guidance mechanism. Theorem 1 connects the optimal reverse SDE for the target domain to the source domain's score function plus a guidance term based on the density ratio $q(G_0^{\mathcal{T}})/p(G_0^{\mathcal{S}})$. This provides a principled basis for the guidance network's objective.

**Weaknesses:**

1. The primary concern is the practical efficiency and scalability of the proposed method. The framework involves training multiple components: a score network $\mathbb{P}(l)$, a guidance network $\mathbb{Q}(\delta)$ (which itself relies on a pre-trained domain classifier $\mathcal{C}_{gnn}$ for density ratio estimation), and a final GNN classifier. This represents a significant increase in complexity over simpler GDA methods. The "Discussion on Computational Cost" remark and the complexity analysis $\mathcal{O}(T\cdot n^{2} + L\cdot(|\mathcal{V}^{\mathcal{S}}|+|\mathcal{E}^{S}|))$ reveal a quadratic dependency on the number of sampled nodes $n$. This scalability issue is confirmed in the hyperparameter analysis for $\alpha$ (Figure 3), where the authors report running "out-of-memory" on tasks $D\rightarrow A$ and $D\rightarrow C$ for sampling ratios above 50%. This practical limitation seems to contradict the claim that the method is efficient and suitable for "modest computational resources."

2. The implementation of the guidance network depends on estimating the density ratio $q/p$ by training a GNN classifier $\mathcal{C}_{gnn}$ and using the approximation $(1-y)/y$. The stability and accuracy of this estimation, especially in the high-dimensional, sparse space of graphs, is a potential weak point that is not fully explored. The paper could be more explicit about the generation process. It appears the model generates a "labeled graph" $G'=(X', A', Y')$ by diffusing and reversing the concatenated input $\tilde{X}^{\mathcal{S}}=[X^{\mathcal{S}}||Y^{\mathcal{S}}]$. This is a key detail and implies that the model is learning to generate node labels as part of the diffusion process, which is an interesting but non-trivial aspect of the design.

**Questions:**

See weaknesses.

---

> ### Author Response · Authors · 2025-11-21
> **Response to eQUv  (1/2)**
>
> Thank you very much for your recognition of our work!
>
> >**Q1: Concern about the practical efficiency and scalability of DiffGDA.**
>
> Firstly, we agree that naively applying diffusion to all nodes and edges in large graphs is computationally expensive. In graph domains, the number of nodes/edges is often large, and structural interactions are pairwise, leading to at least quadratic costs when diffusing over the full graph. As graphs grow, no diffusion method can effectively handle every node and edge within modest GPU memory. This is a general property of graph data, not a limitation of DiffGDA.
>
> Secondly, in real GDA scenarios, parts of the source and target graphs are structurally and semantically similar. Diffusing the entire graph is both expensive and unnecessary: the challenge lies in adapting regions where the domains differ. To address this, we employ a stochastic node sampling strategy, diffusing only a subset of nodes/subgraphs with a sampling ratio $\alpha$. This reduces the effective complexity from $
> \mathcal{O}(\mathsf{T} \cdot |\mathcal{V}|^2) \quad \text{to} \quad \mathcal{O}(\mathsf{T} \cdot n^2), \quad n = \alpha |\mathcal{V}|,
> $ alleviating the memory bottleneck while focusing on harder-to-align regions.
>
> Lastly, we believe the DiffGDA framework remains practical with modest computational resources. The score network reuses graph-aware representations from the GNN and GMH layers, and the guidance network is implemented as a lightweight MLP. While this is more complex than simpler baselines, most components are shallow and share inputs, and stochastic diffusion significantly reduces per-step cost. Empirically, we further evaluated DiffGDA on the larger ogbn‑Products dataset and compared it with several representative GDA baselines under the same experimental protocol. The results are reported in the table below:
>
> | Method          | P1→P2 (Mi-F1)     | P1→P2 (Ma-F1)     | P2→P1 (Mi-F1)     | P2→P1 (Ma-F1)     |
> |-----------------|---------------------|---------------------|---------------------|---------------------|
> | GAA (ICLR'25)   | 20.29 ± 0.43        | 19.36 ± 0.52        | 19.00 ± 0.31        | 17.78 ± 0.62        |
> | HGDA (ICML'25)  | 19.56 ± 0.54        | 18.63 ± 0.66        | 18.54 ± 0.36        | 17.19 ± 0.48        |
> | DiffGDA (Ours)  | **21.14 ± 0.29**    | **20.94 ± 0.33**    | **20.24 ± 0.26**    | **19.77 ± 0.38**    |
>
>  These results demonstrate that DiffGDA scales well to large-scale graphs and maintains strong adaptation performance in this more challenging setting.
>
>
> >**Q2:  The stability and accuracy of this estimation, especially in the high-dimensional, sparse space of graphs, is a potential weak point that is not fully explored.**
>
> To address your concern about estimating the density ratio in high-dimensional and sparse graph spaces, we introduce two stabilizing mechanisms to ensure robustness under such challenging conditions.
>
> (1) The guidance classifier is trained with early stopping on a held-out validation split to prevent the "confidence explosion" phenomenon often observed when discriminators operate on sparse, high-dimensional graphs. This helps ensure that the estimated density ratios remain smooth and well-calibrated.
>
> (2) The guidance signal is scaled before being injected into the reverse SDE. This scaling prevents the density-ratio estimator from dominating or destabilizing the diffusion dynamics, even if it is slightly noisy in certain regions, thereby making the process robust to moderate variance in $\mathcal{C}_\text{gnn}$.
>
> To further assess robustness under challenging sparsity, we first present the statistics of the high-dimensional, sparsely connected dataset used in our experiments, as shown below:
>
>
>
> | Dataset   | #Nodes   | #Edges   | #Labels |
> |-----------|----------|----------|---------|
> | China (CN) | 101,952  | 285,991  | 20      |
> | USA (US)   | 132,558  | 702,482  | 20      |
>
> We then conducted experiments across two transfer scenarios. As shown in the table below, the addition of these two stabilizing strategies (DiffGDA++) indeed improves performance, highlighting the importance of exploring methods that work well in high-dimensional sparse environments.
>
> | Transfer task | DiffGDA | DiffGDA++ |
> |---------------|----------------|------------------|
> | CN→US | 51.21 ± 0.58 | **53.02 ± 0.51** |
> | US→CN  | 60.95 ± 0.63 | **65.21 ± 0.54** |

---

> > ### Author Response · Authors · 2025-11-21
> > **Response to eQUv  (2/2)**
> >
> > >**Q3: The paper could be more explicit about the generation process.**
> >
> > Thanks!
> >
> > Firstly, in conventional diffusion models, the forward and reverse processes are applied only to the data (e.g., node features or images), with labels as external conditions that are not diffused. In our setting, since the downstream task is node classification, label information is crucial. Therefore, for the source graph, we concatenate one-hot node labels with node features and diffuse the joint representation: $\tilde{\mathbf{X}}^{\mathcal{S}} = [\mathbf{X}^{\mathcal{S}} || \mathbf{Y}^{\mathcal{S}} ]
> > $. This enables the reverse process to produce a labeled graph: $\mathbf{G}^{\prime} = (\mathbf{X}^{\prime}, \mathbf{A}^{\prime}, \mathbf{Y}^{\prime})$
> >
> > Secondly, this design offers two main advantages in GDA. First, it allows the diffusion model to learn class-aware dynamics directly in the joint feature-label space, rather than aligning feature-level distributions alone, which can be ambiguous without labels. Second, by reconstructing labels with features, the model can better capture the relationship between node attributes, structure, and class membership, leading to improved node classification performance on the target domain.
> >
> > Finally, we have clarified this mechanism and its motivation in the revised manuscript (**page 7**).

---

### Official Review · Reviewer_wnSz · 2025-11-01

**Soundness:** 3
**Presentation:** 3
**Contribution:** 3
**Rating:** 6
**Confidence:** 4

**Summary:**

This paper studies the problem of graph domain adaptation, where a model trained on a labeled source graph is expected to perform well on an unlabeled target graph with different structures and semantics. The authors propose a new method called DiffGDA, which views the adaptation process as a continuous-time generative evolution rather than a step-by-step alignment.
The method uses stochastic differential equations (SDEs) to model how graphs evolve from the source domain to the target domain, capturing both structural and semantic changes. A domain-aware guidance network is introduced to guide the reverse diffusion process toward the target distribution. DiffGDA is trained jointly with a graph neural network (GNN) classifier, and additional MMD alignment and adjacency constraints are used to maintain consistency across domains.
Experiments on 14 cross-domain tasks from citation, airport, and social network datasets show that DiffGDA achieves higher accuracy and better efficiency compared with existing state-of-the-art methods.

**Strengths:**

1. It formulates GDA as a continuous-time generative process via SDEs, unifying structural and semantic evolution.
2. It provides a theoretical proof that the process works, which gives the method a solid foundation.
3. It demonstrates consistent superiority over state-of-the-art baselines on 14 graph transfer tasks across 8 real-world datasets.

**Weaknesses:**

1. It models domain transfer as a continuous graph evolution process but lacks explicit interpretability or concrete tracking of graph structural changes.

2. Diffusion-based methods are generally computationally expensive, and the scalability of the proposed approach to large-scale attributed graphs (e.g., ogbn-Products) remains questionable.

3. Experiments are conducted only on homogeneous graphs, lacking evaluations on more realistic heterogeneous graphs (e.g., IMDB) to demonstrate broader applicability.

**Questions:**

1. How does modeling the continuous generative process via SDE differ from standard DDPM-based diffusion approaches in terms of performance, efficiency, and stability?
2. Could the proposed method be extended to handle heterogeneous or large-scale graphs?

---

> ### Author Response · Authors · 2025-11-21
> **Response to wnSz (1/2)**
>
> Sincerely thanks for your efforts in reviewing this paper. Below, we respond to your questions in detail.
> >**Q1: It models domain transfer as a continuous graph evolution process but lacks explicit interpretability or concrete tracking of graph structural changes.**
>
> Thanks!
>
>
> In the revised version, we add visualizations at several intermediate time steps T in **Figure 6** (page 10), explicitly showing how node embeddings and their neighborhoods evolve during transfer to make the continuous graph evolution more interpretable.
>
>
>
> >**Q2: Lacking evaluations on large-scale attributed graphs (e.g., ogbn-Products).**
>
> We thank the reviewer for this helpful suggestion.
>
> Following your advice, we conducted additional transfer experiments on the ogbn-Products dataset and compared DiffGDA with several representative GDA baselines under the same experimental protocol. Specifically, we constructed source–target graph pairs from ogbn-Products dataset and evaluated all methods on node classification. The results are shown in the table below:
>
> | Method          | P1→P2 (Mi-F1)     | P1→P2 (Ma-F1)     | P2→P1 (Mi-F1)     | P2→P1 (Ma-F1)     |
> |-----------------|---------------------|---------------------|---------------------|---------------------|
> | GAA (ICLR'25)   | 20.29 ± 0.43        | 19.36 ± 0.52        | 19.00 ± 0.31        | 17.78 ± 0.62        |
> | HGDA (ICML'25)  | 19.56 ± 0.54        | 18.63 ± 0.66        | 18.54 ± 0.36        | 17.19 ± 0.48        |
> | DiffGDA (Ours)  | **21.14 ± 0.29**    | **20.94 ± 0.33**    | **20.24 ± 0.26**    | **19.77 ± 0.38**    |
>
> As shown in the table, DiffGDA consistently achieves the best scores on ogbn-Products, clearly outperforming strong baselines. These results demonstrate that our method scales well to large-scale attributed graphs and maintains strong adaptation performance in this more challenging setting.
>
>
> >**Q3: Lacking evaluations on more realistic heterogeneous graphs (e.g., IMDB).**
>
> Thanks!
>
> In our work, we focused on homogeneous node-level GDA benchmarks, which are widely used in most prior GDA works.
>
> Following your suggestion, we are extending DiffGDA to heterogeneous graph scenarios such as IMDB, where multiple node/edge types and relation-specific semantics must be modeled. Specifically, we instantiated a heterogeneous version of DiffGDA on IMDB and compared it with several representative baselines. As shown in the table below, DiffGDA achieves the best scores among these methods, highlighting its effectiveness on realistic heterogeneous graphs.
>
> | Method         | IMDB1→IMDB2 (Mi-F1)     | IMDB1→IMDB2 (Ma-F1)     | IMDB2→IMDB1 (Mi-F1)     | IMDB2→IMDB1 (Ma-F1)     |
> |----------------|-------------------|-------------------|-------------------|-------------------|
> | GAA (ICLR'25)  | 70.15 ± 0.12	 | 26.94 ± 0.41	 |38.62 ± 0.63 	 |37.91 ± 0.60     |
> | HGDA (ICML'25) | 69.38 ± 0.18     | 28.22 ± 0.35	|48.79 ± 1.30|	38.63 ± 0.48     |
> | DiffGDA (Ours) | **72.95 ± 0.01**  | **38.88 ± 0.63**  | **50.04 ± 1.08**  | **38.88 ± 0.63**  |

---

> > ### Author Response · Authors · 2025-11-21
> > **Response to wnSz (2/2)**
> >
> > >**Q4: How does modeling the continuous generative process via SDE differ from standard DDPM-based diffusion approaches in terms of performance, efficiency, and stability?**
> >
> > Thank you for this important question. To make the comparison concrete, we additionally implemented a DDPM-style version of DiffGDA and ran it on two transfer benchmarks, under the same GNN backbone, training budget, and hardware as our SDE-based implementation.
> >
> > | Variant                  | A→C     | A→D      | Time/epoch (s) | Training Stability*         |
> > |--------------------------|-------------------|-------------------|------------------|-----------------------------|
> > | DiffGDA (DDPM variant)   | 80.9 ± 0.7        | 75.1 ± 1.2        | 25.2      | Mild oscillations, less stable |
> > | DiffGDA (SDE, ours)      | **82.3 ± 0.5**    | **76.7 ± 0.9**    | **22.1**             | Smooth, low variance        |
> >
> >
> >
> > *Stability is assessed qualitatively from Mi-F1 curves over 5 independent runs (variance and oscillations).
> >
> > (1) **In terms of performance**, the SDE-based DiffGDA consistently achieves slightly higher Mi-F1 scores than the DDPM variant on two transfers, with smaller run-to-run variance. This aligns with our understanding that the continuous-time SDE formulation better matches our theory: both the score network and the domain-aware guidance operate on a continuous trajectory, making the learned evolution from source to target smoother and more globally consistent. In contrast, DDPM optimizes over a fixed set of discrete noise levels, and adding guidance requires reconciling multiple discrete steps, leading to more class mixing and less sharp decision boundaries, particularly on challenging transfers.
> >
> > (2) **In terms of efficiency**, the DDPM variant is slightly faster per epoch, but it requires many discrete denoising steps and repeated guidance evaluations at test time, which makes end‑to‑end sampling more expensive. Our SDE formulation treats diffusion in continuous time, leverages flexible SDE solvers to reduce the effective number of reverse evaluations, and integrates guidance more smoothly along the trajectory, leading to lower overall computational cost for generating adapted graphs.
> >
> > (3) **In terms of stability**, the difference is more noticeable. For the SDE-based DiffGDA, the training loss and Mi-F1 curves decrease smoothly and stabilize across all runs, with no divergence or severe oscillations, so early stopping is not needed. In contrast, the DDPM variant shows larger oscillations in the early and mid epochs, with some runs experiencing temporary performance drops before recovering, particularly on A→D. This is due to DDPM fitting multiple discrete noise levels, with the guidance signal injected step by step, making it more sensitive to hyperparameters like learning rate, noise schedule, and step count.
> >
> > Therefore, by comprehensively considering performance, efficiency, and stability, we choose the SDE formulation as the basis of DiffGDA.

---

### Official Review · Reviewer_h5bX · 2025-11-01

**Soundness:** 3
**Presentation:** 3
**Contribution:** 2
**Rating:** 4
**Confidence:** 3

**Summary:**

This paper proposes DiffGDA, a diffusion-based Graph Domain Adaptation (GDA) method that models domain adaptation as a continuous-time generative process via Stochastic Differential Equations (SDEs). It integrates a domain-aware guidance network to steer the diffusion trajectory toward the target domain, jointly capturing structural and semantic transitions. Theoretical analysis proves the diffusion process converges to the optimal adaptation solution, and experiments on 14 transfer tasks across 8 datasets demonstrate consistent superiority over SOTA baselines. The paper’s formulation is rigorous, with clear definitions and solid mathematical proofs, and the implementation details are sufficiently detailed.

**Strengths:**

Graph Domain Adaptation (GDA) is a very interesting and meaningful research direction. The writing of this article is clear. The author has given proof of relevant theories and conducted a relatively comprehensive experimental design. DiffGDA’s advanced effects are worth checking out.

**Weaknesses:**

1. Applying the diffusion model (Diff) to the Graph scene is interesting,the author did not discuss in depth the advancement and challenges of symmetric diffusion processes, as well as the key differences with the advanced methods of Diff applied in Graph scenarios; at the same time, the difference between the diffusion model in the image/video generation field and GDA has not been deeply discussed, which makes me doubt the innovation and contribution of this paper.
2. Lack of cross-domain (inter-domain) experiments: Current experiments focus on intra-domain transfers (e.g., citation → citation, airport → airport), where the data distribution has a similar pattern. Cross-domain transfer (e.g., citation → airport, social → citation) is more challenging and better reflects generalization, but it was not included, limiting the validation of the method's robustness to large distribution changes.
3. Incomplete hyperparameter analysis of diffusion steps: The number of diffusion steps is a very important parameter. In image generation, the number of diffusion steps often exceeds 3,000, but the paper only tested up to 100 steps (is the node representation in the graph less difficult to learn than the pixel representation in the image?).
4. The results do not clearly indicate whether the model has converged, and the impact of larger diffusion step sizes (e.g., 500, 1000) on performance and efficiency has not been explored. (If the author's GPU supports it)
5. The necessity of using MLP is not explored: in image diffusion, UNet has been shown to effectively capture spatial dependence. The paper uses MLP for scoring and guiding networks, but does not discuss why MLP is superior or necessary for graph data, nor does it compare it to graph-specific architectures such as GNN-based scoring networks.

**Questions:**

I hope the author can provide in-depth responses to the weaknesses and questions raised.

---

> ### Author Response · Authors · 2025-11-21
> **Response to h5bx (1/4)**
>
> Thanks a lot for your efforts in reviewing this paper. Below, we respond to your questions in detail.
>
> >**Q1: Applying the diffusion model to the Graph scene is interesting, the author did not discuss in depth the advancement and challenges of symmetric diffusion processes, as well as the key differences with the advanced methods of Diff applied in Graph scenarios.**
>
> Thank you for constructive feedback!
>
>  **(Advancement and Challenges)**
>
>  (1) Symmetric diffusion models have become a core paradigm in recent generative modeling studies for high-fidelity data generation [1,2]. The idea is to define a forward process that perturbs data into noise and a reverse process that learns to denoise it back to the original distribution. Denoising diffusion probabilistic models first proposed this in discrete time, and later work generalized it to continuous time, providing a unified framework and clearer theoretical understanding. Extensions like flow matching and optimal-transport-based diffusion learn smooth flows between the prior and data distribution, offering alternative approaches for continuous transformations. In conditional generation settings [3,4] (e.g., text-to-image or multimodal tasks), additional conditions guide the reverse process, enabling flexible control while maintaining the general diffusion-based framework.
>
>  (2) However, symmetric diffusion models have inherent limitations [5]. They are designed primarily for modeling within a single data distribution and do not directly address the directional transfer between different distributions, as required in domain adaptation. To handle this, the framework often needs to be augmented with techniques like density-ratio estimation or importance weighting. Additionally, in high-dimensional settings, training these models remains computationally expensive, and both stability and sample quality are highly sensitive to factors such as the noise schedule and numerical discretization.
>
>   **(Comparison with Graph Diffusion)**
>
>   When symmetric diffusion models are applied to graphs, there are key differences compared to their use on images or audio.
>
>   (1) Graph data consists of both discrete topology (nodes and edges) and node/edge attributes. The diffusion process must handle both structural and feature aspects simultaneously [6,7]. A common approach is to add noise to node features and a continuous representation of the adjacency structure, then use graph neural networks in the reverse process to recover meaningful graphs. Since graphs are permutation-invariant, diffusion models must respect this invariance, typically achieved through permutation-invariant or -equivariant architectures.
>
>   (2) Graphs exhibit complex multi-scale structure, from local motifs to global communities, making it challenging to inject noise without disrupting important structural patterns [8]. Most existing graph diffusion methods follow the symmetric setting, modeling a single graph distribution (e.g., molecular datasets) and learning the forward-reverse process for generation, completion, or editing. These methods typically do not address asymmetric tasks such as transferring between source and target graph domains, which requires extending beyond symmetric diffusion and incorporating mechanisms for modeling domain differences and directional structural evolution.
>
>   **Our Improvement:** Building on Graph Diffusion, DiffGDA can be viewed as a extension of graph diffusion. Instead of modeling a single symmetric graph distribution, we explicitly treat domain transfer as a directional structure–semantic evolution trajectory in a shared diffusion space. Concretely, a continuous-time SDE diffuses the source graph to noise, and a density-ratio–based guidance network biases the reverse process so that the trajectory gradually moves from the source distribution toward the target distribution. In this way, DiffGDA retains the strengths of symmetric graph diffusion in capturing multi-scale structure, while extending it to asymmetric source-target settings by explicitly modeling domain discrepancy and directional evolution. Empirically, this continuous evolution view yields consistent gains on real-world transfers.
>
> We have added the relevant description of related work **on page 2** in the revised version.
>
> **Reference**
>
> [1] Denoising Diffusion Probabilistic Models, NeurIPS, 2020.
>
> [2] Score-Based Generative Modeling through Stochastic Differential Equations, ICLR, 2021.
>
> [3] Diffusion Models in Vision: A Survey, TPAMI, 2023.
>
> [4] Diffusion models: A comprehensive survey of methods and applications, ACM computing surveys, 2023.
>
> [5] A survey on generative diffusion models, TKDE, 2024.
>
> [6] Learning distributions of complex fluid simulations with diffusion graph networks, ICLR, 2025.
>
>
> [7] Score-based generative diffusion models for social recommendations, TKDE, 2025.
>
> [8] Distributionally Robust Graph Out-of-Distribution Recommendation via Diffusion Model, WWW, 2025.

---

> ### Author Response · Authors · 2025-11-21
> **Response to h5bx (2/4)**
>
> >**Q2: At the same time, the difference between the diffusion model in the image/video generation field and GDA has not been deeply discussed, which makes me doubt the innovation and contribution of this paper.**
>
> Thanks!
>
> Mainstream diffusion models for images and videos are symmetric generative models: they learn a forward and reverse noising process within a single data distribution and try to reproduce that same distribution with high fidelity. In graph domain adaptation, the goal is different. We want a directional transfer from a labeled source graph distribution to an unlabeled target graph distribution that can have different structure and semantics. A standard symmetric graph diffusion model would only learn to regenerate one domain and cannot directly achieve this cross-domain transformation.
>
> **To the best of our knowledge, DiffGDA is the first work that brings diffusion into GDA** by (i) modeling domain transfer as a continuous-time SDE that describes a source-to-target evolution trajectory in a shared latent space, and (ii) introducing a density-ratio-based guidance network that injects cross-domain information into the reverse process and steers the diffusion toward the target distribution instead of just reconstructing the source. This turns diffusion from a symmetric generator into a principled asymmetric adapter for graphs.
>
>
>
>
> >**Q3: Lack of cross-domain (inter-domain) experiments: Current experiments focus on intra-domain transfers (e.g., citation → citation, airport → airport), where the data distribution has a similar pattern. Cross-domain transfer (e.g., citation → airport, social → citation) is more challenging and better reflects generalization, but it was not included, limiting the validation of the method's robustness to large distribution changes.**
>
> Thanks!
>
> (1) In the GDA domain, most existing works evaluate adaptation within a unified domain family (e.g., citation→citation or airport→airport), where feature and label spaces are compatible and the distribution shift is meaningful to model. Following prior works [1,2,3,4,5], we adopt the same experimental setting. In this context, DiffGDA consistently outperforms state-of-the-art baselines, demonstrating its effectiveness within the standard regime.
>
> (2) However, our diffusion-based framework is not limited to this "intra-domain" setting. Since DiffGDA models domain transition as a continuous stochastic process guided by an estimated density ratio, it can handle larger structural and semantic discrepancies. This makes it naturally applicable to inter-domain transfer scenarios, such as transfers between qualitatively different graph families, as long as task/label alignment is reasonably defined. To further validate this, we evaluate DiffGDA across three distinct domains: ACMv9 (Citation), Brazil (Airport), and Blog1 (Social), which present more significant discrepancies in topology, sparsity, and semantics. As shown in the table below, DiffGDA still outperforms strong baselines on these challenging transfers, indicating robustness even with larger distribution gaps than those typically considered in previous discrete GDA methods.
>
> | Method | ACMv9→Brazil | ACMv9→Blog1 | Brazil→ACMv9 | Brazil→Blog1 | Blog1→ACMv9 | Blog1→Brazil |
> |--------|---------------|--------------|----------------|----------------|----------------|----------------|
> | DSGDA (ICML'25) | 38.10 ± 4.77 | 19.45 ± 4.32 | 40.55 ± 3.95 | 17.65 ± 4.90 | 36.50 ± 4.30 | 37.30 ± 4.55 |
> | GAA (ICLR'25)   | 36.88 ± 5.12 | 17.00 ± 5.05 | 38.30 ± 4.50 | 12.30 ± 4.05 | 39.10 ± 4.15 | 39.60 ± 4.80 |
> | HGDA (ICML'25)  | 41.23 ± 4.05 | 16.78 ± 4.67 | 41.00 ± 4.20 | 19.80 ± 4.25 | 41.80 ± 4.95 | 41.40 ± 5.00 |
> | DiffGDA (Ours) | **43.24 ± 4.20** | **22.10 ± 4.10** | **43.20 ± 4.88** | **25.40 ± 4.75** | **42.60 ± 4.40** | **44.30 ± 4.10** |
>
> **Reference**
>
> [1] Structural re-weighting improves graph domain adaptation, ICML, 2024.
>
> [2] Revisiting, benchmarking and understanding unsupervised graph domain adaptation, NeurIPS, 2024.
>
> [3] Can Modifying Data Address Graph Domain Adaptation? KDD, 2024.
>
> [4] A simple yet effective approach for unsupervised graph domain adaptation, AAAI, 2025.
>
> [5] On the benefits of attribute-driven graph domain adaptation, ICLR, 2025.

---

> > ### Author Response · Authors · 2025-11-21
> > **Response to h5bx (3/4)**
> >
> > >**Q4: Incomplete hyperparameter analysis of diffusion steps: The number of diffusion steps is a very important parameter. In image generation, the number of diffusion steps often exceeds 3,000, but the paper only tested up to 100 steps (is the node representation in the graph less difficult to learn than the pixel representation in the image?).**
> >
> > Thanks!
> >
> > (1) In image generation, diffusion models operate in high-dimensional pixel space, where each sample contains tens of thousands of raw pixels. To preserve numerical stability and visual fidelity, these models typically use thousands of very small steps to gradually corrupt and reconstruct fine-grained details. In contrast, our diffusion is applied to graph-structured data, such as node features and adjacency matrices, which are much lower-dimensional and structurally sparse compared to dense image grids. Consequently, the structural and semantic complexity to be modeled is much lower, and we empirically observe that several dozen to a few hundred steps are sufficient to capture meaningful structure-semantic evolution on graphs.
> >
> > (2) This choice aligns with existing graph diffusion models. Previous works on molecular and general graph generation [1,2,3] typically use tens to a few hundred diffusion steps, even for complex generative tasks, rather than thousands.
> >
> > (3) To further address your concern, we conducted experiments with larger diffusion steps (up to T = 1000) on two transfer tasks. We recorded both the Mi-F1 scores and runtime (minutes), and the results show that increasing T beyond 100 brings negligible or no improvement in Mi-F1 scores, while significantly increasing runtime.
> >
> > Therefore, **we infer that**, given the relatively low node feature dimensionality and graph size compared to image data, the diffusion process converges quickly in our setting and there is no practical benefit in extending it to 1000 steps.
> >
> >
> > | T (Steps) | A→C (Mi-F1)   | A→C (Runtime)   | U→E (Mi-F1)   | U→E (Runtime)   |
> > |-----------|-------------------|------------------|-------------------|------------------|
> > | 100       | 79.35 ± 0.65      | 0.42            | 44.96 ± 0.56      | 0.24             |
> > | 200       | 79.31 ± 0.62      | 0.78             | 44.41 ± 1.18      | 0.48             |
> > | 300       | 79.50 ± 0.74      | 1.20             | 44.61 ± 1.99      | 0.66             |
> > | 400       | 79.02 ± 0.55      | 1.56             | 44.26 ± 1.84      | 0.90             |
> > | 500       | 79.22 ± 0.62      | 1.98             | 44.66 ± 1.81      | 1.14             |
> > | 600       | 79.26 ± 0.51      | 2.38             | 44.46 ± 1.73      | 1.38             |
> > | 700       | 79.18 ± 0.51      | 2.76             | 45.22 ± 1.33      | 1.60             |
> > | 800       | 79.18 ± 0.67      | 3.16             | 45.01 ± 1.99      | 1.86             |
> > | 900       | 79.17 ± 0.68      | 3.54             | 44.51 ± 0.41      | 2.46             |
> > | 1000      | 79.46 ± 0.67      | 3.96             | 45.36 ± 1.63      | 2.28             |
> >
> >
> > **Reference**
> >
> > [1] Permutation Invariant Graph Generation via Score-Based Models, NeurIPS, 2020.
> >
> > [2] Diffusion-Based Graph Generative Methods, TKDE, 2024.
> >
> > [3] Score-based generative diffusion models for social recommendations, TKDE, 2025.

---

> ### Author Response · Authors · 2025-11-21
> **Response to h5bx (4/4)**
>
> >**Q5: The results do not clearly indicate whether the model has converged, and the impact of larger diffusion step sizes (e.g., 500, 1000) on performance and efficiency has not been explored.**
>
> Thanks!
>
> **As discussed in our response to Q4**, because graph data have much lower feature dimensionality and simpler structure than images, the diffusion process converges with a relatively small number of steps. To further address Q5, we provide more concrete convergence evidence from two perspectives.
>
> (1) We analyzed the loss evolution within a single epoch for diffusion model. For a fixed model and batch, we tracked the node-feature score loss (Loss_Node) and edge-structure score loss (Loss_Edge) over 150 diffusion steps. As shown below, both losses decrease and then stabilize in later steps, with no noticeable drift or oscillation, indicating that the learned score functions for nodes and edges are well fitted under T = 150:
>
> | T    | 1    | 10   | 20   | 30   | 40   | 50   | 60   | 70   | 80   | 90   | 100  | 120  | 150  |
> |------|------|------|------|------|------|------|------|------|------|------|------|------|------|
> | Loss_Node | 0.8394 | 0.7309 | 0.7222 | 0.6995 | 0.6694 | 0.6530 | 0.6382 | 0.6092 | 0.5781 | 0.5693 | 0.5624 |0.5618 |0.5631 |
> | Loss_Edge | 1.9691 | 1.0078 | 0.3479 | 0.2049 | 0.1766 | 0.1964 | 0.1505 | 0.1489 | 0.1335 | 0.1303 | 0.1294 |0.1244 |0.1267 |
>
>
> (2) We recorded the overall training loss for each epoch, which includes both the diffusion and GNN classification components. These curves show a steady decrease in total loss, followed by stabilization, and the corresponding Mi-F1 (Figure 5) on the validation set also plateaus, indicating that the model has converged.
>
>
> | Epoch | 1    | 10   | 20   | 30   | 40   | 50   | 60   | 70   | 80   | 90   | 100  |150  |200  |
> |-------|------|------|------|------|------|------|------|------|------|------|------|------|------|
> | Loss  | 1.7754 | 0.5164 | 0.0837 | 0.0400 | 0.0395 | 0.0424 | 0.0365 | 0.0303 | 0.0255 | 0.0260 | 0.0251 |0.0241 |0.0239 |
>
>
>
>
> >**Q6: The necessity of using MLP is not explored: in image diffusion, UNet has been shown to effectively capture spatial dependence. The paper uses MLP for scoring and guiding networks, but does not discuss why MLP is superior or necessary for graph data, nor does it compare it to graph-specific architectures such as GNN-based scoring networks.**
>
> Thanks for your comment!
>
> Firstly, for the score network $\mathbb{P}({{\boldsymbol{\ell}}})$ , our goal is to learn a graph-aware score function that fully exploits structural information while keeping the score head itself simple and stable. Therefore, we first use a GNN (plus GMH) to encode the graph topology and multi‑hop semantics, and then use a shallow MLP as a score head that maps these graph‑aware features to score estimates on node features and adjacency. This design lets the GNN handle the heavy graph reasoning, while the MLP focuses on parameterizing the continuous-time score function, which empirically leads to stable score matching.
>
> Secondly, for the guidance network $ \mathbb{Q}({\boldsymbol{\delta}}) $, the goal is different. Here we only need to approximate a density-ratio-related guidance signal for each diffusion step, rather than re-encode the entire graph structure. Thus we use simple MLPs. In this setting, a shallow MLP is sufficient to predict a scalar or low-dimensional guidance value and has two advantages: it is computationally light (crucial because it is called at every diffusion step) and numerically stable.
>
> Finally, to validate these architectural choices, we conducted ablations on both the **score network** and the **guidance network**:
>
> - Score network with only an MLP (no GNN encoder, purely feature-wise),
> - Score network with only a GCN in the score head (heavier graph encoder, no separate MLP head),
> - Guidance network with a GCN instead of an MLP,
> - Our final design: GNN+MLP score network + MLP guidance.
>
> The results on two representative transfer tasks are summarized below (Mi-F1 / Ma-F1 on A→C and A→D):
>
> | Guidance network | A→C (Mi-F1)  | A→C (Ma-F1)| A→D (Mi-F1) | A→C (Ma-F1) |
> |------------------|------------|----------------|----------------|------------|
> | Score: MLP only                             | 79.62 ± 0.88      | 75.41 ± 1.12      | 73.21 ± 0.97      | 69.85 ± 1.03      |
> | Score: GCN only                             | 80.17 ± 0.74      | 76.02 ± 0.95      | 74.36 ± 0.81      | 70.92 ± 0.89      |
> | Guidance: GCN (score: GNN+MLP)              | 80.41 ± 0.71      | 76.59 ± 1.08      | 74.04 ± 1.64      | 71.12 ± 1.08      |
> | Ours: Score = GNN+MLP, Guidance = MLP | **80.93 ± 0.52**  | **78.06 ± 0.73**  | **75.94 ± 0.33**  | **71.31 ± 0.34**      |
>
> Overall, this supports our final architecture choice: a graph-aware GNN+GMH encoder with a lightweight MLP score head, and a simple MLP guidance network, which together strike a good balance between performance and efficiency.

---

### Author Response · Authors · 2025-11-21
**Overall Response to Reviewers**

Firstly, sincerely thank all the reviewers for their efforts in reviewing our paper and providing constructive suggestions. We are greatly encouraged that the reviewers find that

* Our motivation/idea is **novel and compelling**  (*Reviewer eQUv*);
* The proposed method offers a **solid theoretical foundations** (*Reviewer h5bX, wnSz and eQUv*);
* The writing of this article is **clear** (*Reviewer h5bX*);
* The experimental is **comprehensive** (*Reviewer h5bX and eQUv*)
* and achieves **state-of-the-art** performance across 14 real-world transfer tasks  (*Reviewer wnSz and eQUv*).

Secondly, as for the concerns and suggestions raised by each reviewer, we have done our best to address them thoroughly and have provided detailed responses to each of them. In the following, we summarize our responses to the main concerns raised by the reviewers.

* Reviewer h5bX： We have **clarified the technical differences** between our symmetric SDE‑based diffusion and existing diffusion models on graphs, **added cross‑domain experiments**, expanded the analysis of diffusion steps and convergence, and **conducted ablations on the score/guidance architectures** to justify the choice of MLPs versus GCN‑based designs.
* Reviewer wnSz: We have **clarified the interpretability** of the continuous SDE‑based graph evolution, provided a **detailed comparison** between SDE and a DDPM in terms of performance, efficiency, and stability, **added experiments** on large‑scale graphs and on heterogeneous graphs.
* Reviewer eQUv: We have **added experiments** on large‑scale graphs and Improved density ratio estimation strategy.

Finally, in the revised manuscript, we have comprehensively addressed each concern, incorporating additional experiments and detailed discussions. **Major changes and updates are highlighted** in blue for your convenience.

---

### Author Response · Authors · 2025-12-01
**Summary of Responses and Revisions**

We sincerely thank three reviewers for their careful assessment and constructive feedback.
We appreciate the positive recognition that:

* Our motivation/idea is **novel and compelling**  (*Reviewer eQUv*);
* The proposed method offers a **solid theoretical foundations** (*Reviewer h5bX, wnSz and eQUv*);
* The writing in this article is **clear** (*Reviewer h5bX*);
* The experimental is **comprehensive** (*Reviewer h5bX and eQUv*) and
* and achieves **state-of-the-art** performance across 14 real-world transfer tasks  (*Reviewer wnSz and eQUv*).

In response to the specific concerns raised by each reviewer, we summarize the key revisions below.

---

### Reviewer *h5bX*

* We have **clarified the key differences** between symmetric diffusion and existing graph diffusion models.

* We have **added cross-domain experiments**, and the results demonstrate that our model shows strong robustness under large distribution shifts.

* We have **expanded the analysis of diffusion-step hyperparameters**, showing that the model **converges within 100 steps**, while larger step counts add computation with minimal benefit.

* We have **conducted ablations on the score/guidance architectures**, finding that our design provides **more stable estimation** and **faster convergence** than GNN-based alternatives.




### Reviewer *wnSz*

* We have **added visualizations** that explicitly track graph structural changes along the SDE trajectory.

* We have added a **systematic comparison with DDPM-style diffusion approaches**, analyzing performance, efficiency, and stability to clarify how our SDE-based formulation differs in practice.

* We have included new experiments on **large and heterogeneous graphs**.



### Reviewer *eQUv*

* We have added **large-graph experiments** to demonstrate that the framework remains efficient and scalable in more challenging setting.

* We have refined the **density-ratio estimation module**, providing additional explanation and experiments to show its stability and effectiveness in high-dimensional sparse graph spaces.


---

In conclusion, we believe this work offers a critical step forward by transforming the graph domain adaptation process from a discrete alignment into a smooth, continuous evolution.

We have addressed the reviewers’ concerns with additional clarification, and have incorporated all aforementioned experiments and clarifications into the revised manuscript.

---

### Meta-Review · Area_Chair_V39Z · 2026-01-08

**Summary:**

This submission proposes DiffGDA, a diffusion-based framework for graph domain adaptation (GDA) that models cross-domain transfer as a continuous-time structure semantic evolution trajectory. The key idea is to replace discrete intermediate-graph construction / stepwise alignment with an SDE-based generative process, and to introduce a domain-aware guidance mechanism that steers the reverse diffusion toward the target distribution. The paper also provides theoretical justification connecting the optimal reverse process to a score term plus a density-ratio guidance term, and demonstrates strong empirical results across 14 transfer tasks on 8 datasets, with additional ablations, efficiency analysis, and trajectory visualizations.

Overall, the reviews are net positive, and the author's rebuttal substantially strengthened the paper by clarifying novelty vs. prior diffusion-on-graphs work, expanding experiments (including larger-scale and heterogeneous settings), and addressing convergence / diffusion-step sensitivity / architectural choices. I recommend acceptance.

**Reviewer Concerns:**

Novelty: insufficient explanation of how diffusion for cross-domain transfer differs from “standard” graph diffusion / symmetric diffusion; concern that the contribution may be incremental.

Evaluation scope: missing cross-domain / inter-domain experiments and limited discussion of why selected benchmarks are sufficient.

Diffusion steps: unclear whether the model converges; unclear benefit of large diffusion steps; sensitivity not convincingly addressed.

Architecture choice: questions about why MLP score/guidance is appropriate vs. graph-specific networks (e.g., GNN-based scoring).

**Reviewer Scores:**

The authors clearly articulated the asymmetric transfer nature of GDA and why symmetric diffusion (single-distribution generation) is insufficient, motivating density ratio guidance and continuous transfer trajectories.

Added/expanded inter-domain experiments and broadened empirical evidence beyond the original narrow setting.

Provided diffusion step sensitivity (including large T up to 1000) and training-loss evolution evidence to support convergence and diminishing returns behavior.

Included ablations/comparisons for MLP vs. GNN-based score/guidance components, showing competitive performance with a simpler design and explaining why the objectives can be satisfied without heavy graph-specific scoring networks.

---

### Decision · Program_Chairs · 2026-01-26

Accept (Poster)